# Stepwise nucleosome translocation by RSC remodeling complexes

**Bryan T Harada[1,2], William L Hwang[1,2,3], Sebastian Deindl[2,4†], Nilanjana Chatterjee[5‡], Blaine Bartholomew[5], Xiaowei Zhuang[2,4,6*]**

[1]Graduate Program in Biophysics, Harvard University, Cambridge, United States; [2]Howard Hughes Medical Institute, Harvard University, Cambridge, United States; [3]Harvard/MIT MD-PhD Program, Harvard Medical School, Boston, United States; [4]Department of Chemistry and Chemical Biology, Harvard University, Cambridge, United States; [5]Department of Epigenetics and Molecular Carcinogenesis, University of Texas M.D. Anderson Cancer Center, Smithville, United States; [6]Department of Physics, Harvard University, Cambridge, United States

*For correspondence: zhuang@chemistry.harvard.edu

Present address: †Department of Cell and Molecular Biology, Uppsala University, Uppsala, Sweden; ‡School of Medicine, University of California, San Francisco, San Francisco, United States

**Abstract** The SWI/SNF-family remodelers regulate chromatin structure by coupling the free energy from ATP hydrolysis to the repositioning and restructuring of nucleosomes, but how the ATPase activity of these enzymes drives the motion of DNA across the nucleosome remains unclear. Here, we used single-molecule FRET to monitor the remodeling of mononucleosomes by the yeast SWI/SNF remodeler, RSC. We observed that RSC primarily translocates DNA around the nucleosome without substantial displacement of the H2A-H2B dimer. At the sites where DNA enters and exits the nucleosome, the DNA moves largely along or near its canonical wrapping path. The translocation of DNA occurs in a stepwise manner, and at both sites where DNA enters and exits the nucleosome, the step size distributions exhibit a peak at approximately 1–2 bp. These results suggest that the movement of DNA across the nucleosome is likely coupled directly to DNA translocation by the ATPase at its binding site inside the nucleosome.

## Introduction

The SWI/SNF-family chromatin remodelers are important regulators of chromatin structure in transcriptional activation and in the creation and maintenance of nucleosome-free regions (*Clapier and Cairns, 2009*; *Bowman, 2010*; *Becker and Workman, 2013*; *Narlikar et al., 2013*; *Bartholomew, 2014*; *Lorch and Kornberg, 2015*). Consistent with these roles, SWI/SNF remodelers are capable of disrupting nucleosomes in a number of ways, including repositioning the histone octamer along the DNA, ejecting H2A-H2B dimers from the octamer, and ejecting the histone octamer from the DNA (*Clapier and Cairns, 2009*; *Bowman, 2010*; *Becker and Workman, 2013*; *Narlikar et al., 2013*; *Bartholomew, 2014*; *Lorch and Kornberg, 2015*). All SWI/SNF complexes are composed of a catalytic subunit, which harbors a superfamily 2 (SF2) ATPase domain, and a variety of accessory subunits (*Clapier and Cairns, 2009*). During remodeling, the ATPase domain contacts the nucleosome at an internal site 20 bp from the nucleosome dyad, referred to as the super-helical 2 (SHL2) site, and translocates DNA (*Saha et al., 2002*; *Saha et al., 2005*; *Zofall et al., 2006*; *Dechassa et al., 2008*). However, how DNA translocation by the ATPase domain is coupled to the various activities of the remodeling complexes remains unclear.

SWI/SNF remodelers expose substantial amounts of intra-nucleosomal DNA to nucleases during remodeling, suggesting that the remodelers disrupt the wrapping of DNA around the nucleosome (*Narlikar et al., 2001*; *Aoyagi et al., 2002*; *Lorch et al., 2010*; *Shukla et al., 2010*). The nature of

**eLife digest** Cells package their genetic information in a "complex" of proteins and DNA called chromatin. This complex is made of units called nucleosomes, each of which consist of a short stretch of DNA wrapped around proteins known as histones. These nucleosomes restrict access to the DNA wrapped around the histone proteins, and thus serve to regulate whether genes are activated and a variety of other cellular processes. Certain enzymes regulate the structure of chromatin by altering the position and structure of nucleosomes. However, it is not clear exactly how these "chromatin remodeling" enzymes alter the contacts between the DNA and histone proteins to move DNA around the nucleosome.

RSC is a chromatin-remodeling enzyme that typically helps to activate genes. Harada et al. used a technique called single molecule fluorescence resonance energy transfer (or single molecule FRET for short) to observe the movement of DNA around the histone proteins. The technique involves placing a green fluorescent dye on the histone proteins and a red fluorescent dye on the DNA. If the red dye is close to the green dye, some of the energy can be transferred from the green dye to the red dye when the green dye is excited by a laser. By looking at the ratio of green and red light emitted, it is possible to tell how far apart they are, and how this changes over time.

The experiments show that the RSC enzyme moves the DNA into and out of the nucleosome in small steps. These steps match the expected step size of DNA movements by a section of the enzyme called the ATPase domain. This suggests that the ATPase domain drives the motion of DNA across the entire nucleosome. A future challenge is to better understand how chromatin remodeling enzymes cooperate with other molecules in cells to remodel nucleosomes and chromatin.

the disrupted intermediates remains unclear, and previous studies have suggested unpeeling of DNA from the edges of the nucleosome (*Bazett-Jones et al., 1999*; *Floer et al., 2010*) as well as the formation of loops or bulges of DNA inside the nucleosome (*Narlikar et al., 2001*; *Kassabov et al., 2003*; *Zhang et al., 2006*; *Chaban et al., 2008*; *Shukla et al., 2010*; *Liu et al., 2011*; *Ramachandran et al., 2015*).

Given that the ATPase domain of the enzymes acts at the SHL2 site, approximately 50 and 100 bp away from the sites where the DNA enters and exits the nucleosome, respectively, how ATPase activity drives the overall movement of DNA around the nucleosome remains unclear. The isolated catalytic subunit of a SWI/SNF remodeler has been shown to translocate naked DNA (in the absence of a histone octamer) in ~2 bp steps (*Sirinakis et al., 2011*). However, DNA translocation around the nucleosome by SWI/SNF holoenzyme complexes has been reported to occur in large steps of ~50 bp in size (*Zofall et al., 2006*). It is unknown whether this 50-bp step represents the fundamental step size of nucleosome translocation by SWI/SNF complexes. The related ISWI-family of chromatin remodelers move DNA in multi-bp compound steps that are composed of 1-bp fundamental steps (*Blosser et al., 2009*; *Deindl et al., 2013*). Although the ISWI-family remodelers share a homologous ATPase domain with the SWI/SNF-family remodelers, they differ in both the domains flanking the ATPase domain on their catalytic subunits and the accessory subunits that associate with the catalytic subunits (*Clapier and Cairns, 2009*), and they bind the nucleosome differently (*Hota and Bartholomew, 2011*; *Dechassa et al., 2012*). Thus, it is unclear whether SWI/SNF remodelers share a similar nucleosome translocation mechanism.

In this study, we used single-molecule FRET (*Ha et al., 1996*) with a variety of labeling schemes to monitor nucleosome remodeling by yeast RSC, a prototypical SWI/SNF-family enzyme (*Cairns et al., 1996*), in real time. These experiments allowed us to observe transient intermediates of the remodeling process and characterize the motion of the DNA at the nucleosomal edges where the DNA moves into and out of the nucleosome. At these locations, we found that DNA was translocated largely along or near its canonical path in a stepwise manner and that the distribution of the step sizes showed a peak at a step size of approximately 1–2 bp.

## Results

### Single-molecule FRET assay for monitoring nucleosome translocation by RSC

We reconstituted dye-labeled mononucleosomes on a double-stranded DNA containing the 601 positioning sequence (*Lowary and Widom, 1998*) to ensure reproducible positioning of the histone octamer. We used several labeling schemes with the FRET donor dye on various sites of the histone octamer and the FRET acceptor dye on various locations of the DNA. In the first labeling scheme, the mononucleosome was flanked by a shorter 6 bp linker on one side and a longer 78 bp linker on the other side. The FRET acceptor dye, Cy5, was attached to the end of the shorter linker and a biotin moiety was attached to the end of the longer linker for immobilization on the microscope slide (*Figure 1A*). The histone octamer was labeled with the FRET donor dye, Cy3, on the C-terminal tail of H2A (position 119). We refer to this construct as the H2A/[end, +6] construct to indicate the position of the Cy3 label (on histone *H2A*) and Cy5 label (at the *end* of the linker DNA, *6 bp* from the edge of the nucleosome). This labeling scheme allowed us to monitor the movement of DNA relative to the histone octamer in real time during nucleosome remodeling.

We anticipate that remodeling of the dye-labeled mononucleosomes by RSC will generate two distinct products, depending on which side of the nucleosome the ATPase engages (*Figure 1A*). If RSC engages the SHL+2 site, which we define to be the SHL2 site near the longer linker DNA, RSC will translocate DNA toward the shorter linker. Previous studies of SWI/SNF-family remodelers have shown that the enzyme can translocate DNA around the nucleosome until the end of the DNA reaches the SHL2 site, which is ~50 bp past the edge of the nucleosome (*Flaus and Owen-Hughes, 2003*; *Kassabov et al., 2003*). Therefore, this type of action should generate a ~130 bp movement of the DNA toward the shorter linker, moving the Cy5 dye away from the Cy3 dye on the octamer and causing a monotonic decrease in FRET. This action will eventually position the Cy5-labeled DNA end >40 nm from the Cy3 label on the H2A, resulting in zero FRET. On the other hand, if RSC engages the SHL−2 site, the SHL2 site near the shorter linker DNA, RSC will translocate DNA toward the longer linker, first moving the Cy5 dye closer to the Cy3 dye on the octamer and then further away from the Cy3, causing an initial increase in FRET followed by a decrease. This action will generate a final product where the labeled DNA end resides at the SHL−2 site. Based on the crystal structure of the nucleosome (*Luger et al., 1997*), this should place the Cy5 ~6.8 nm from the Cy3 labeling site, giving a low but non-zero FRET value.

Consistent with these expectations, we observed two major classes of single-molecule traces upon addition of RSC and ATP to the nucleosomes. One class of traces showed a monotonic decrease in FRET to zero FRET (*Figure 1B*). We assigned these traces to the case where the ATPase domain of RSC bound to the SHL+2 site and translocated the DNA toward the shorter linker. Because the dye labels monitored the dynamics of the DNA end moving away from the octamer, we refer to these traces as monitoring exit-side movement. The second class of traces showed an initial increase in FRET followed by a decrease to a final FRET of ~0.17 (*Figure 1C*). We assigned these traces to the case where the ATPase domain bound to the SHL−2 site and translocated DNA toward the longer linker, which is expected to first bring the FRET donor and acceptor dyes closer and then move them farther apart. Because the dye labels in these cases monitored the dynamics of the DNA end moving into the nucleosome, we refer to these traces as monitoring entry-side movement. We classified the traces as reflecting entry-side or exit-side movement based on the presence or absence of an initial FRET increase during remodeling and identified a roughly equal number traces showing entry-side and exit-side movement when RSC and ATP were added to the H2A/[end, +6] nucleosome construct (*Figure 1D*). This result is consistent with previous results indicating that translocation by SWI/SNF enzymes is bidirectional and does not depend on linker DNA length (*Flaus and Owen-Hughes, 2003*; *Kassabov et al., 2003*; *Shundrovsky et al., 2006*). It has been shown previously that the strength of the histone-DNA contacts between the 601 positioning sequence and the histone octamer is asymmetric with respect to the nucleosomal dyad (*Ngo et al., 2015*), so we tested whether the fraction of nucleosomes undergoing exit-side or entry-side movement depends on the orientation of the 601 positioning sequence by reversing the 601 sequence on the H2A/[end, +6] nucleosome. We found that the fractions of traces exhibiting entry-side and exit-side movement were essentially identical for the two sequence orientations (*Figure 1—figure supplement 1*), suggesting that the asymmetry of the 601 positioning sequence does not influence the directionality of

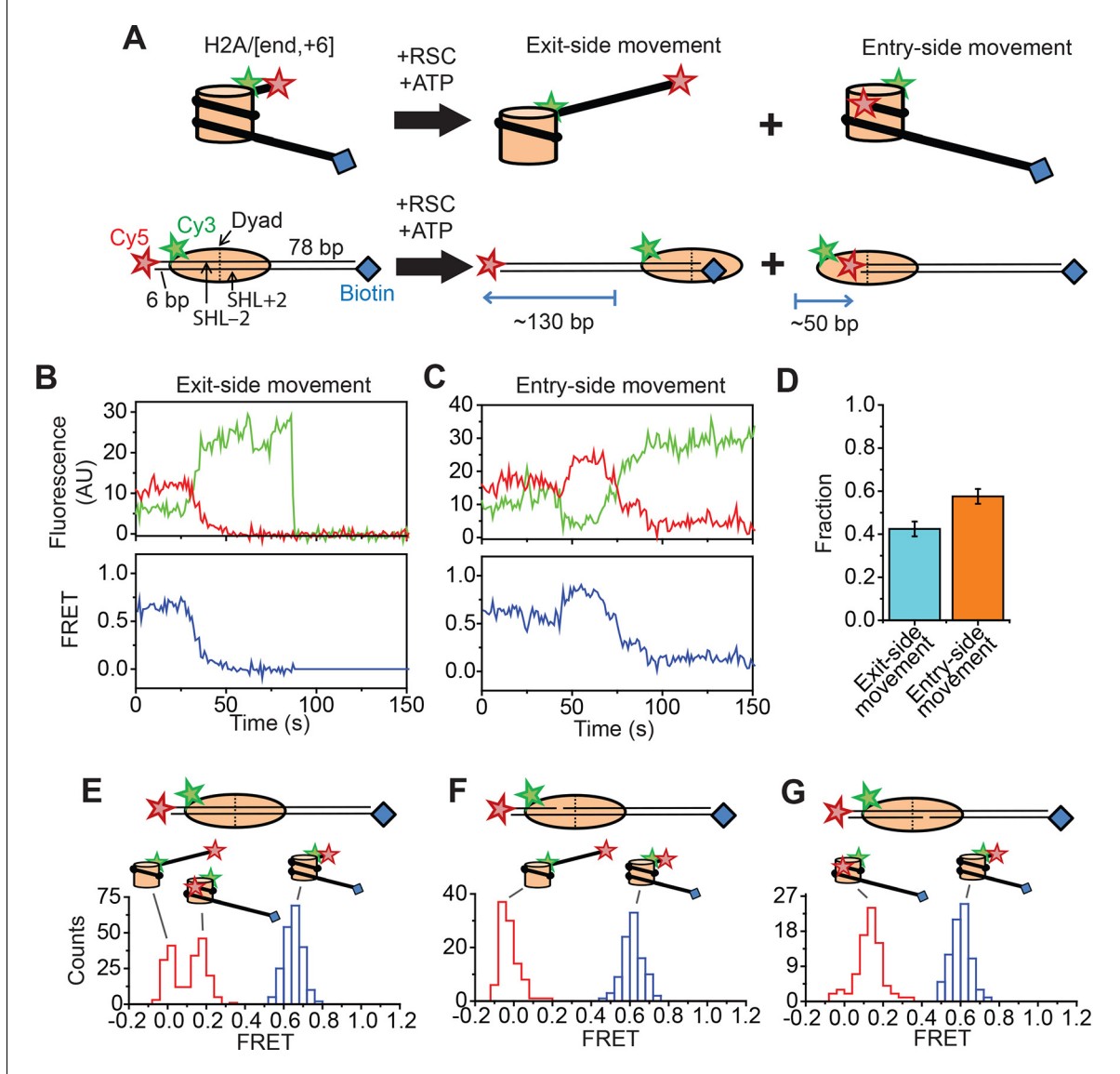

**Figure 1.** Single-molecule FRET assay for monitoring nucleosome translocation by RSC. (A) Diagram depicting the nucleosome substrates before and after remodeling by RSC. The top row depicts the nucleosomes in cartoon form while the bottom row shows the footprint of the histone octamer (tan oval) on the DNA (black lines). (B and C) Representative traces showing the Cy3 intensity (green), Cy5 intensity (red), and FRET value (blue) for exit-side movement (B) and entry-side movement (C). [RSC] = 5 nM, [ATP] = 5 μM. (D) The fraction of exit-side movement (cyan) and entry-side movement (orange) traces observed with 1 nM RSC and 20 μM ATP. Error bars represent the standard error from > 200 nucleosomes. (E–G) Histograms of the distribution of initial (before remodeling, blue) and final (after remodeling, red) FRET values of remodeling traces from nucleosomes lacking any ssDNA gap (E), nucleosomes with a 2-nt ssDNA gap at the SHL–2 site (F), and nucleosomes with a 2-nt ssDNA gap at the SHL+2 site (G). [RSC] = 1 nM, [ATP] = 20 μM.

The following figure supplements are available for figure 1:

**Figure supplement 1.** Remodeling of nucleosomes with a reversed 601 positioning sequence.

**Figure supplement 2.** Surface anchoring of the nucleosomes does not affect the kinetics of remodeling.

**Figure supplement 3.** Locations of the 2-nt ssDNA gaps in the DNA sequence.

**Figure supplement 4.** Effects of 2-nt ssDNA gaps at the SHL ± 2 sites on the direction of nucleosome translocation.

remodeling by RSC. The overall kinetics of the FRET changes observed in the single-molecule assay were similar to the kinetics observed in solution-based ensemble FRET measurements, indicating that surface attachment did not substantially affect remodeling by RSC (*Figure 1—figure supplement 2*). As expected, no major FRET changes were observed when RSC was added to the nucleosomes in the absence of ATP (*Figure 1—figure supplement 2*).

In order to confirm our assignment of entry-side and exit-side movement traces, we made use of the fact that the ATPase of RSC is incapable of translocating past a 2-nt single-stranded (ss) gap in the DNA (*Saha et al., 2005*; *Zofall et al., 2006*). Therefore, placing such a gap at the SHL+2 or SHL–2 site (*Figure 1—figure supplement 3*) should allow us to control the direction of DNA translocation around the nucleosome. After remodeling by RSC, nucleosomes without any ssDNA gap showed two distinct populations of nucleosomes with FRET values of ~0 and ~0.17, corresponding to the products of exit-side and entry-side movement, respectively (*Figure 1E*). As expected, remodeling of a construct containing a gap at the SHL–2 site showed only a single peak at zero FRET after remodeling (*Figure 1F*), consistent with the gap preventing RSC from engaging the SHL–2 site to generate entry-side movement traces. Furthermore, >95% of the remodeling traces from this construct were classified as exit-side movement traces and showed a monotonic FRET decrease without an initial FRET increase (*Figure 1—figure supplement 4A,B*). Similarly, a gap at the SHL+2 site largely eliminated the zero FRET population and resulted in essentially a single population with a FRET of ~0.14 after remodeling (*Figure 1G*). Only a small fraction (~8%) of nucleosomes showed a final FRET of zero, which is most likely due to photobleaching of the Cy5 dye during remodeling. Moreover, >95% of the traces observed were classified as entry-side movement traces and showed an initial FRET increase during remodeling (*Figure 1—figure supplement 4C,D*), consistent with the gap preventing RSC from engaging the SHL+2 site and generating exit-side movement traces. These results confirm that the traces classified as reflecting entry-side movement resulted from ATPase action at the SHL–2 site and the traces classified as reflecting exit-side movement resulted from ATPase action at the SHL+2 site.

## Probing for potential H2A-H2B dimer movement during remodeling

In addition to nucleosome sliding, RSC has been reported to facilitate a number of other changes to the nucleosome, such as ejecting H2A-H2B dimers or the entire histone octamer, but we do not expect to observe these activities in our assay because these activities require free nucleosomes or free DNA as acceptors, the activity of histone chaperones, or a dinucleosome construct (*Clapier and Cairns, 2009*; *Bowman, 2010*; *Becker and Workman, 2013*; *Narlikar et al., 2013*; *Bartholomew, 2014*; *Lorch and Kornberg, 2015*). Indeed, after incubating the H2A/[end,+6] nucleosomes with 5 nM RSC and 5 µM ATP for 20 min, >90% of the nucleosomes remained present with the H2A subunit attached, similar to the fraction of nucleosomes remaining intact after incubation in the absence of RSC and ATP (91%). However, because RSC could destabilize the H2A-H2B dimer (*Lorch et al., 2010*), it is possible that RSC may cause transient H2A-H2B dimer displacement during remodeling that contributes to the FRET changes. To investigate whether movement of the H2A-H2B dimer (and hence the Cy3-label on H2A) relative to the rest of the octamer contributed to the FRET changes we observed during RSC remodeling, we took three approaches. First, we created a H3/[end,+6] construct (*Figure 2A*) by moving the Cy3 label from the C-terminal tail of histone H2A to the N-terminal tail of histone H3 (position 33). If movement of the H2A-H2B dimer relative to the rest of the octamer was responsible for the FRET changes that we observed on the H2A/[end,+6] construct, we expected that moving the Cy3 dye to histone H3 would eliminate these FRET changes. However, if H2A-H2B dimer dynamics did not contribute the observed FRET changes, the H3-labeled nucleosomes would produce FRET dynamics similar to those observed with the H2A-labeled nucleosomes because the N-terminal tail of histone H3 lies near the C-terminal tail of histone H2A (*Luger et al., 1997*). Similar to H2A-labeled nucleosomes, remodeling of the H3-labeled nucleosomes also produced two classes of FRET traces, one class showing a monotonic decrease in FRET (*Figure 2B*) and the other class showing an initial FRET increase followed by a FRET decrease (*Figure 2C*). As in the case of the H2A/[end,+6] constructs, placing a gap at the SHL–2 site of the H3/[end,+6] nucleosomes eliminated nearly all of the traces showing an initial FRET increase, and placing a gap at the SHL+2 site of the H3/[end,+6] nucleosomes eliminated most of the traces showing a monotonic FRET decrease (*Figure 2D*), indicating that these traces reflect entry-side and exit-side movement, respectively. The observation that the FRET changes were similar for the H2A-

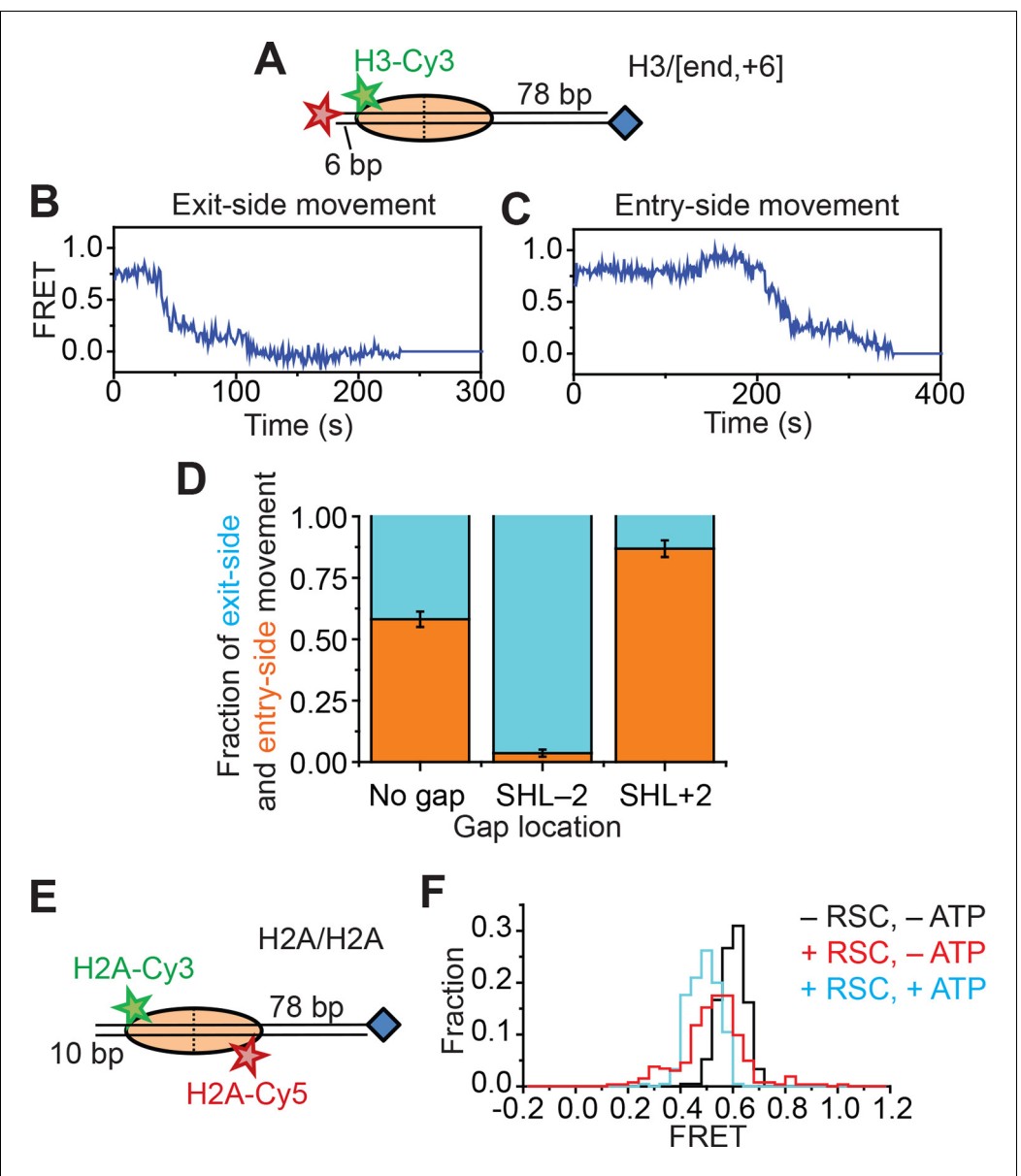

**Figure 2.** Single-molecule FRET assay for probing the displacement of the H2A-H2B dimer during RSC-mediated nucleosome remodeling. (**A**) Diagram of the H3/[end,+6] nucleosome. (**B** and **C**) Representative FRET traces reflecting the exit-side movement (**B**) and entry-side movement (**C**) of the H3/[end,+6] nucleosomes. [RSC] = 1 nM, [ATP] = 2 μM. (**D**) Fraction of traces showing entry-side movement (orange) and exit-side movement (cyan) observed with 1 nM RSC and 20 μM ATP for nucleosomes lacking any ssDNA gap, nucleosomes with a 2-nt ssDNA gap at the SHL−2 site, and nucleosomes with a 2-nt ssDNA gap at the SHL+2 site. Error bars represent the standard error from > 100 nucleosomes per construct. (**E**) Diagram of the H2A/H2A nucleosomes. (**F**) Histogram of the FRET values from the H2A/H2A nucleosomes in the absence of RSC (black), after the addition of 6 nM RSC (red), or after the addition of 6 nM RSC and 5 μM ATP (cyan). Histograms were constructed from >200 nucleosomes per condition.

The following figure supplements are available for figure 2:

**Figure supplement 1.** Remodeling of the H2A/H2A construct by RSC.

**Figure supplement 2.** Remodeling of nucleosomes labeled in the globular domain of histone H2B.

labeled and H3-labeled nucleosomes suggests that these observed FRET changes were not specific to the dynamics of the H2A-H2B dimer.

Second, to directly probe for potential movement of the H2A-H2B dimer during remodeling, we reconstituted histone octamers with a 1:1 mixture of Cy3-labeled H2A and Cy5-labeled H2A and assembled mononucleosomes using these octamers and unlabeled DNA. We referred to these nucleosomes as H2A/H2A to indicate that both donor and acceptor dyes are on the H2A subunits (*Figure 2E*). Nucleosomes with a single Cy3 on one of the H2A subunits and a single Cy5 on the other H2A subunit could be readily identified at the single-molecule level and gave FRET ~ 0.6 (*Figure 2F*), whereas nucleosomes lacking a Cy5 showed zero FRET and nucleosomes lacking a Cy3 were not visible under green laser illumination. Addition of RSC and ATP to this H2A/H2A construct resulted in only a small decrease in FRET, ΔFRET ~0.1 (*Figure 2F* and *Figure 2—figure supplement 1A*). This FRET decrease was partially recapitulated by the addition of enzyme in the absence of ATP (*Figure 2F*), suggesting that this FRET change was associated, at least in part, with RSC binding. Furthermore, this apparent FRET change was due almost entirely to an increase in Cy3 fluorescence without a corresponding decrease in Cy5 fluorescence (*Figure 2—figure supplement 1A,B*), suggesting that this apparent FRET change was probably due to a change in the photophysical properties of the Cy3 dye upon RSC binding rather than a *bona fide* change in distance between the dyes. Even if the FRET change was in part due to a dye-to-dye distance change, the small magnitude of this change was not sufficient to explain the much larger changes in FRET observed with the H2A/[end,+6] and H3/[end,+6] constructs during remodeling.

Finally, because the H2A labeling site resides on the flexible, basic tail of this histone subunit, it is possible that this region could interact with the negatively-charged DNA and mask the detection of H2A-H2B motion. To address this possibility, we moved the dye label to histone H2B at position 49, which resides within the globular domain of the histone. We first reconstituted a H2B/H2B nucleosomal construct containing a mixture of Cy3- and Cy5-labeled H2B on unlabeled DNA, as we did for the H2A/H2A construct. We identified nucleosomes containing a Cy3 dye on one H2B subunit and a Cy5 dye on the other H2B subunit by selecting those nucleosomes with a finite FRET value (FRET~0.2), again because nucleosomes lacking a Cy5 showed zero FRET and nucleosomes lacking a Cy3 were not visible under green laser illumination. The FRET values of these H2B/H2B nucleosomes did not change appreciably after addition of RSC and ATP (*Figure 2—figure supplement 2A*). Next, to observe the remodeling behavior of the nucleosomes with H2B-labeled octamer, we constructed a H2B/[end,+0] nucleosome, containing Cy3 on the *H2B* subunit (position 49) and a Cy5 at the 5' *end* of the DNA, positioned at the edge of the nucleosome (the *+0* position) (*Figure 2—figure supplement 2B*). Upon addition of RSC and ATP to the H2B/[end,+0] nucleosomes, we again observed both exit-side and entry-side movement traces (*Figure 2—figure supplement 2C,D*) resembling those obtained with the H2A- and H3-labeled nucleosomes. The H2B/[end,+0] nucleosomes showed a preference for entry-side remodeling (*Figure 2—figure supplement 2E*), perhaps because the label biased binding of RSC in the orientation supporting entry-side remodeling. However, control constructs with gaps at the SHL–2 or SHL+2 site eliminated nearly all of the entry-side or exit-side movement traces, respectively (*Figure 2—figure supplement 2E*), confirming our assignment of entry-side and exit-side movement traces.

Thus, in our assays using nucleosomes labeled with Cy3 on the octamer and Cy5 on the DNA, we saw similar entry-side and exit-side movement traces when using three different positions of the Cy3 on the octamer (on H2A, H3 or H2B), suggesting that the FRET dynamics we observed were due primarily to movement of the DNA relative to the octamer. Furthermore, when we placed both Cy3 and Cy5 dyes on the H2A-H2B dimer, we did not observe substantial motion of the H2A-H2B dimer with two different positions of the dye (on H2A or H2B). However, our data cannot exclude the possibility of smaller-scale movement of the H2A-H2B dimer, not detectable by our FRET assay, that could disrupt important histone-histone or histone-DNA contacts during remodeling.

## Probing for potential DNA unwrapping at the edge of the nucleosome

Next, we asked if RSC induces large-amplitude unwrapping of DNA at the edges of the nucleosome. We separately considered this possibility for the two edges of the nucleosome where DNA enters or exits the nucleosome. At the nucleosomal edge where DNA enters the nucleosome, translocation of the DNA along its canonical path on the nucleosome would be expected to produce a FRET increase as the Cy5 at the end of linker DNA moves toward the edge of nucleosome, followed by a FRET

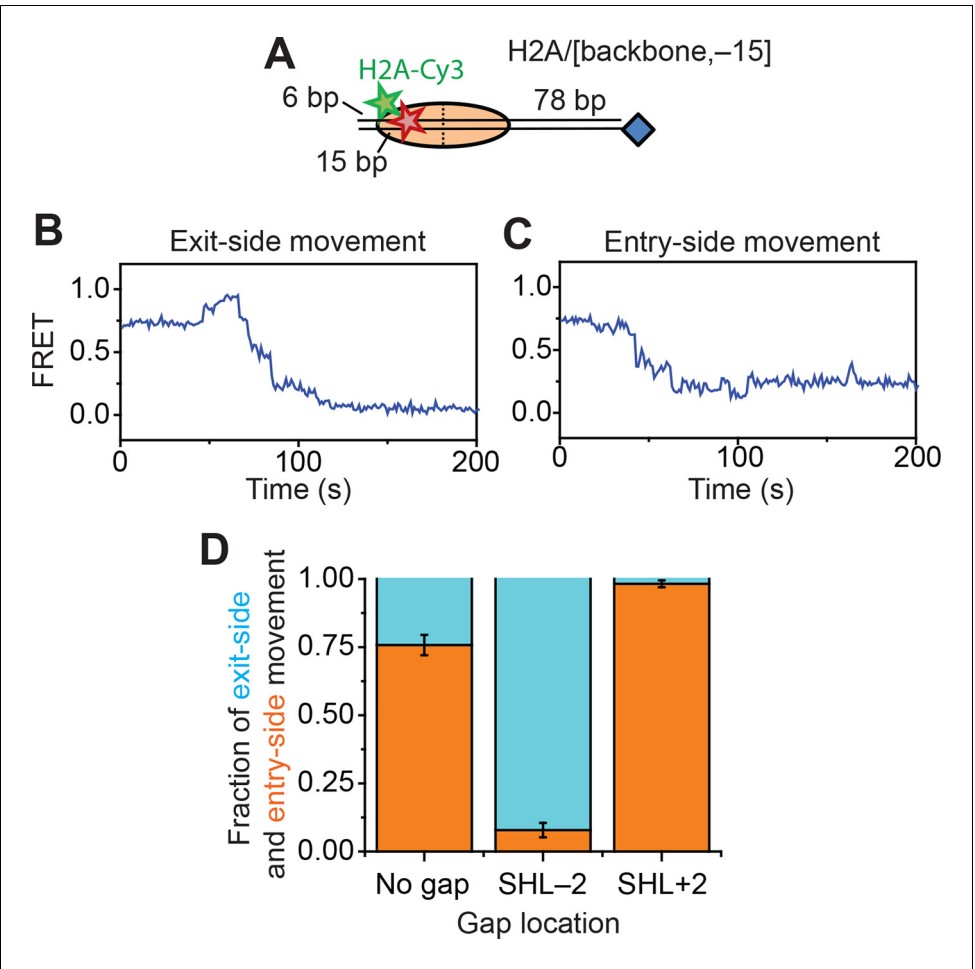

**Figure 3.** Assay for probing DNA unwrapping at the nucleosomal edge where DNA exits the nucleosome. (**A**) Diagram of the H2A/[backbone,–15] nucleosome construct. (**B** and **C**) Representative FRET traces showing exit-side movement (**B**) and entry-side movement (**C**). [RSC] = 1 nM, [ATP] = 5 µM. (**D**) The fraction of entry-side movement (orange) and exit-side movement (cyan) traces observed with 1 nM RSC and 20 µM ATP for nucleosomes lacking any ssDNA gap, nucleosomes with a 2-nt ssDNA gap at the SHL–2 site, and nucleosomes with a 2-nt ssDNA gap at the SHL+2 site. Error bars represent the standard error from >100 nucleosomes per construct.

The following figure supplement is available for figure 3:

**Figure supplement 1.** Assays for probing DNA unwrapping at the nucleosomal edge where DNA enters the nucleosome.

decrease as the Cy5 moves along the nucleosome surface toward the SHL–2 site. On the other hand, if RSC were to unwrap a substantial amount of DNA and lift the DNA off the nucleosomal surface by a large distance, as expected if the unwrapped DNA extended from the nucleosome in a unbent fashion, we would expect the FRET traces to exhibit a FRET decrease as the Cy5 dye on the linker DNA moves away from the nucleosome, where the Cy3 dye resides. The entry-side movement traces that we observed showed a substantial increase in FRET followed by a FRET decrease for both H2A/[end, +6] and H3/[end, +6] constructs (*Figures 1C*, *2C*, *Figure 1—figure supplement 4C* and *Figure 3—figure supplement 1A*), which was consistent with movement largely along or near the canonical path at the nucleosome edge where DNA enters the nucleosome. This phenomenon was also seen for constructs with increased linker DNA length, such as the H2A/[end,+11] and H3/[end,+9] constructs, where the Cy5 dye-labeled DNA end was initially 11 bp and 9 bp away from the nucleosome edge (*Figure 3—figure supplement 1B,C*). As described above, similar entry-side

movement traces were also observed for the H2B/[end,+0] construct, in which the Cy3 label resides inside the globular domain of histone H2B instead of on the flexible tail of histone H2A (*Figure 2—figure supplement 2D*). Some of these entry-side movement traces also showed a small apparent decrease in FRET (ΔFRET ~ 0.1) prior to the FRET increase, but this apparent FRET decrease resulted primarily from an increase in Cy3 intensity without a corresponding decrease in Cy5 intensity (*Figure 3—figure supplement 1A-F*). The magnitude of this Cy3 intensity change was consistent with the changes seen when we labeled the two H2A subunits on the octamer with Cy3 and Cy5 (i.e. the H2A/H2A construct) (*Figure 3—figure supplement 1G*), suggesting that this FRET change likely resulted from a change in the photophysical properties of the Cy3 dye upon RSC binding. Even if these initial FRET decreases reflected a real change in distance between the Cy3 and Cy5 dyes, the magnitude of the FRET change, ΔFRET ~ 0.1 (*Figure 3—figure supplement 1D-F*), was much smaller than both the FRET changes that we observed for RSC-induced DNA translocation around the nucleosome and the FRET changes previously observed for transcription factor-mediated DNA unwrapping from the edge of the nucleosome (ΔFRET ~ 0.6) on a similarly labeled nucleosome construct (*Li and Widom, 2004*). Assuming a previously measured Förster radius of 6 nm for the Cy3-Cy5 pair (*Murphy et al., 2004*), these small initial FRET decreases that we observed would correspond to a change in the Cy3-Cy5 distance of only ~0.5 nm. Thus, our data suggest that under our remodeling conditions, the DNA was not lifted by a large distance away from the nucleosomal surface and did not deviate substantially from its canonical wrapping path as it entered the nucleosome.

In order to determine whether the DNA was unwrapped and lifted by a large distance off the nucleosomal surface at the nucleosomal edge where DNA exits, our current labeling scheme was not adequate since both lifting of DNA off the nucleosomal surface and translocation of DNA along the canonical path on the nucleosome would generate a decrease in FRET. We therefore moved the Cy5 label from the end of the DNA to a site 15 bp inside the edge of nucleosomes by incorporating the dye into the sugar-phosphate backbone to generate H2A/[backbone, −15] nucleosomes, where H2A again indicates the position of the Cy3 label and [backbone, −15] indicates the position of the Cy5 label (*Figure 3A*). If RSC translocates the DNA around the nucleosome along its canonical path, the exit-side movement traces should show an initial FRET increase as the Cy5 moves toward the nucleosome edge, followed by a FRET decrease as the Cy5 exits the nucleosome. In contrast, lifting the DNA by a large distance off the nucleosome surface should produce a substantial FRET decrease, as this structural change would move the Cy5 dye on the DNA farther away from the Cy3 dye on the octamer, assuming that the unwrapped DNA remains largely unbent. Addition of RSC and ATP to this construct again generated two classes of traces: exit-side movement traces showing a transient increase in FRET followed by a decrease to zero FRET (*Figure 3B*) and entry-side movement traces showing a decrease in FRET to ~0.23 (*Figure 3C*). Without any ssDNA gap on the nucleosome, the FRET traces of the H2A/[backbone, −15] construct preferentially showed entry-side movement (*Figure 3D*), likely because placing the Cy5 inside the nucleosome biased RSC to bind in the orientation that positioned the ATPase at the SHL–2 site. Nevertheless, when we introduced a 2-nt ssDNA gap at the SHL–2 site, nearly all of the traces showed the exit-side movement behavior, displaying an initial increase in FRET followed the decrease to zero FRET; when a 2-nt ssDNA gap was introduced at the SHL+2 site, nearly all of the traces showed the entry-side movement behavior, displaying only a monotonic decrease in FRET (*Figure 3D*). These experiments confirm that the traces showing the initial FRET increase followed by a FRET decrease resulted from ATPase action at the SHL+2 site and thus represented exit-side movement. Because lifting of the DNA by a large distance away from the nucleosomal surface would have caused a FRET decrease instead of an initial FRET increase, our data suggest that the DNA was not lifted away from the nucleosome surface by a large distance and did not deviate substantially from its canonical path as it exited the nucleosome.

Although our data are not consistent with RSC moving the DNA at the nucleosomal edges by a large distance away from the octamer surface, as would be expected if RSC were to unwrap a substantial amount of DNA from the edge and allow the DNA to follow its unconstrained path, our data do not exclude the possibility that the enzyme disrupts many of the histone-DNA contacts simultaneously during remodeling (*Lorch et al., 2010*), but still holds the unwrapped or lifted DNA near the surface of the nucleosome.

## Characterization of the step size of DNA translocation

Next, we characterized the step size with which RSC translocated DNA into and out of the nucleosome. To determine the step size of DNA translocation at the nucleosomal edge where DNA exits the nucleosome, we first monitored remodeling of the H2A/[end,+6] nucleosomes at 20°C (as opposed to 30°C in the previous experiments) with 2 μM ATP in order to slow the remodeling reaction and better resolve the translocation steps. The exit-side movement traces exhibited intermittent pauses interrupting the monotonic decrease to zero FRET (*Figure 4A*), and we applied a step-finding algorithm based on chi-square minimization (*Kerssemakers et al., 2006*) to identify the location of the pauses and the sizes of the steps. The step size histogram showed a distribution of step sizes with a major peak centered at a ΔFRET of 0.1 and a tail extending to larger ΔFRET values (*Figure 4B*). Using an alternative, hidden Markov model (HMM)-based algorithm (*McKinney et al., 2006*), we identified similar steps in the FRET traces and produced a similar step size histogram, again with the major peak centered at a ΔFRET of 0.1 (*Figure 4—figure supplement 1*). We note that our results cannot completely exclude the possibility that a small fraction of nucleosomes undergo large-step movement that brings the FRET to zero in a single step as these traces would not be distinguishable from Cy5 photobleaching. However, during remodeling of the H2A/[backbone,–15] construct (*Figure 3*), where exit-side movement led to an increase in FRET that can be distinguished from Cy5 photobleaching, the vast majority (85%) of the exit-side movement traces showed gradual FRET changes inconsistent with such large step sizes, suggesting that at most a small fraction of nucleosomes could remodel with such large-step movement.

Next, we determined the step sizes of DNA translocation assuming that the DNA moved along its canonical path around the nucleosome. To calibrate the FRET changes associated with moving the DNA along its canonical path, we positioned Cy5 at the end of the linker DNA and varied the linker DNA length (*Figure 4—figure supplement 2A*, blue histograms). Consistent with previous results (*Blosser et al., 2009*), these measurements showed an approximately linear relationship between the linker DNA length and the observed FRET value over the measured range with a slope of $0.053 \pm 0.004$ bp$^{-1}$ (*Figure 4—figure supplement 2B*, blue). Because the photophysical effects associated with enzyme binding may affect this relationship, we also performed the calibration on enzyme-bound nucleosomes by measuring the FRET value of the first observed step during remodeling of these constructs (*Figure 4—figure supplement 2A*, red histograms) as a function of the initial linker DNA length. These experiments also showed a linear relationship with a similar slope of $0.047 \pm 0.005$ bp$^{-1}$ (*Figure 4—figure supplement 2B*, red). From these slopes, we determined that the major step size of DNA movement, corresponding to the peak of the ΔFRET distribution in *Figure 4B* (ΔFRET ~ 0.1), to be ~2 bp.

The step size histogram, however, showed many steps that were larger than expected for ~2 bp of DNA motion (*Figure 4B*). Although these data could reflect a heterogeneous step size, the distribution could also result from movement with a single step size, where the larger steps represented the sum of two or more steps where the intervening pause(s) were too short to detect. To test whether our data were consistent with such a model, we took advantage of the linear dependence of FRET on the linker DNA length and fit our data to a series of evenly spaced Gaussian peaks whose amplitude decreased by a factor of *f* from the previous peak, representing the probability of missing a step (see Materials and methods). The data derived from the step-finding algorithm based on chi-square minimization are consistent with a step size of ΔFRET = $0.104 \pm 0.002$ with a probability of missing a step being *f* = 34% (*Figure 4B* and *Figure 4—figure supplement 3A*). Analyzing the step size distribution generated from the alternative HMM-based step-finding algorithm produced a step size of ΔFRET of $0.102 \pm 0.003$ with a missed fraction of *f* = 41% (*Figure 4—figure supplement 1B*). Given the similar results from the two analysis methods, we used only the chi-square-minimization-based step-finding algorithm in subsequent analyses. The probabilities of missing a step obtained from these fits are reasonable given that the steps occur stochastically, and many pauses could be too short to be observed. Based on the distribution of observed pause durations, we would expect a 25% probability of missing a step due to our limited time resolution (*Figure 4—figure supplement 3B*). The observed probability of missing a step was moderately larger than that expected from our time resolution limitation, suggesting the possibility of additional mechanisms for missing steps, for example, successive DNA translocation steps generated by the ATPase at the SHL2 site merging together while transiting to the edge of the nucleosome.

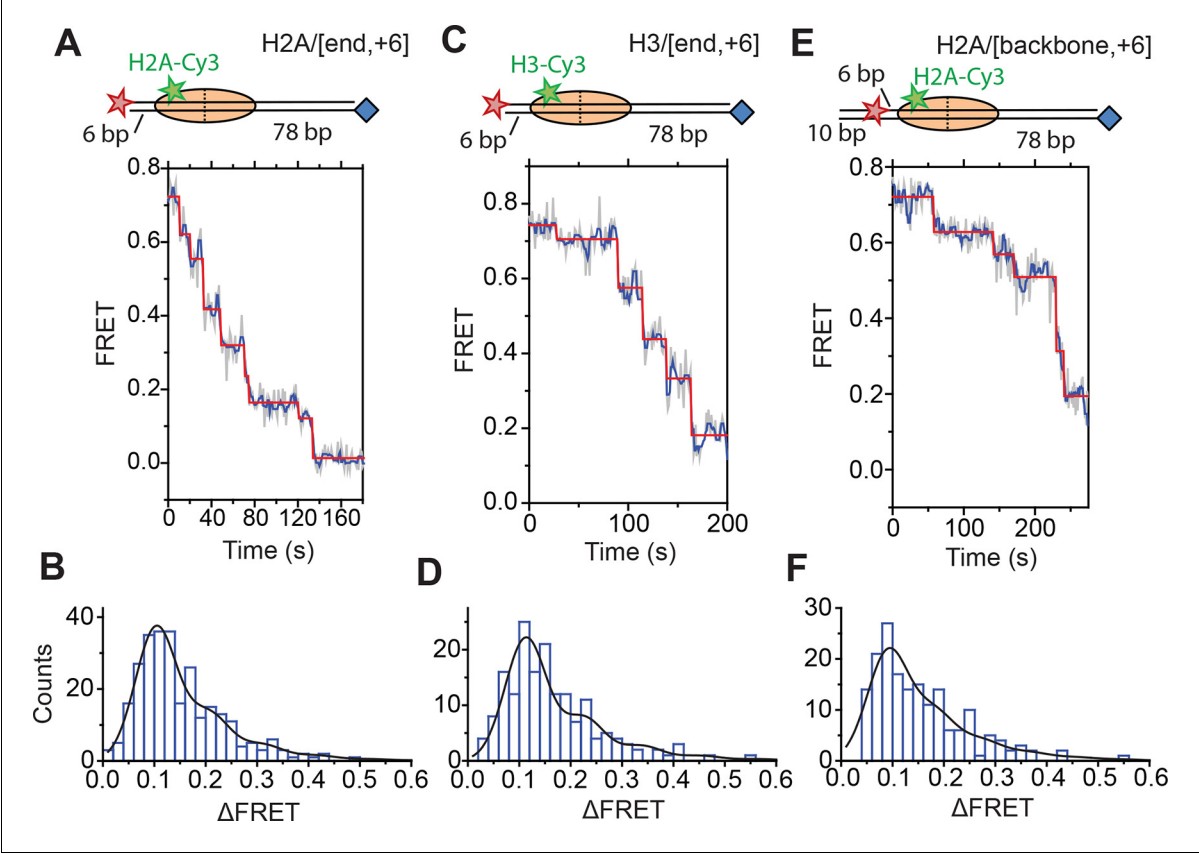

**Figure 4.** DNA exits the nucleosome in a stepwise manner, exhibiting a step size distribution peaked at ~1–2 bp during RSC-mediated remodeling. Remodeling was monitored for H2A/[end,+6] (**A** and **B**), H3/[end,+6] (**C** and **D**), and H2A/[backbone,+6] (**E** and **F**) nucleosome constructs. (**A, C,** and **E**) Top: diagram of the nucleosome construct used. Bottom: Representative exit-side movement traces in the presence of 5 nM RSC and 2 μM ATP at 20°C. Light grey, raw FRET data; blue, 5-point median-filtered data; red, fit by a step-finding algorithm based on Chi-square minimization. (**B, D,** and **F**) Histograms of the measured step sizes in FRET change (blue bars) and the fit to the modeled step size distribution shown in *Equation 1* (black line).

The following figure supplements are available for figure 4:

**Figure supplement 1.** Step size determination using a hidden Markov model (HMM)-based step-finding algorithm.

**Figure supplement 2.** Calibration of FRET values as a function of the linker DNA length for constructs monitoring exit-side movement.

**Figure supplement 3.** Analysis of DNA translocation step sizes of exit-side movement.

**Figure supplement 4.** Monitoring exit-side DNA motion at higher ATP concentrations.

To test whether the step sizes are sensitive to ATP concentration, we determined the step size of the enzyme at two additional concentrations of ATP, 20 μM and 80 μM by performing the experiments with a faster camera frame rate and increased laser intensities (*Figure 4—figure supplement 4*). The FRET from the exit-side movement traces again decreased in a stepwise manner, and the step size distributions exhibited a major peak at a ΔFRET of ~0.1 with a tail extending to larger values (*Figure 4—figure supplement 4*). Fitting the resulting step size histograms to multiple Gaussian peaks as described earlier indicated that the distributions are consistent with a step size of 1.7 ± 0.2 bp with a missed step probability of $f$ = 26% for 20 μM ATP and 2.0 ± 0.2 bp with a missed step probability of $f$ = 41% for 80 μM ATP (*Figure 4—figure supplement 3A,B*). These step size estimates at higher ATP concentrations are similar to the results observed at 2 μM ATP.

To test whether the step size determination was sensitive to the labeling scheme, we monitored remodeling (at 2 μM ATP) with a second nucleosome construct where we moved the Cy3 dye from

histone H2A to histone H3 (the H3/[end, +6] construct, *Figure 4C,D*) and a third construct where we moved the Cy5 to the middle of the linker DNA on the DNA backbone (instead of at the end of the linker DNA) 6 bp from the nucleosome edge (the H2A/[backbone, +6] construct, *Figure 4E,F*). Calibration curves for these two labeling schemes showed that the slopes of the FRET versus linker DNA length were $0.059 \pm 0.005$ bp$^{-1}$ and $0.057 \pm 0.002$ bp$^{-1}$, respectively (*Figure 4—figure supplement 2C-F*, blue). Like for the H2A/[end, +6] construct, these values did not change appreciably in the presence of a bound RSC enzyme (*Figure 4—figure supplement 2C-F*, red). The exit-side movement traces of these new constructs also showed stepwise DNA translocation (*Figure 4C,E*), and the step size distributions (*Figure 4D,F*) showed a major peak around a ΔFRET of 0.1 with a tail extending to larger values. Fitting these histograms to multiple Gaussian peaks as described earlier indicated that the distributions were consistent with a step size of ΔFRET = $0.113 \pm 0.003$ with a missed step probability of $f = 34\%$ for H3/[end, +6] and ΔFRET = $0.092 \pm 0.004$ and $f = 42\%$ for H2A/[backbone,+6] (*Figure 4—figure supplement 3A,C*). Comparing the ΔFRET values to the slopes of the calibration curves gave step size estimates of $1.9 \pm 0.2$ bp and $1.6 \pm 0.1$ bp for the H3/[end, +6] and H2A/[backbone, +6] constructs, respectively, consistent with the $2.0 \pm 0.2$ bp step size estimate for the H2A/[end, +6] construct. Again, because some of the estimated missed step probabilities were moderately larger than would be expected based on our time resolution (*Figure 4—figure supplement 3*), the DNA may have occasionally exited the nucleosome with larger step sizes.

Next, we characterized the step size of DNA translocation at the nucleosomal edge where DNA enters the nucleosome. In the experiments that characterized the step sizes of exit-side motion, the dye-labeled linker DNA was "upstream" (to the left) of the 601 sequence (*Figure 4A,C,E*), so exit-side movement traces resulted from ATPase action at the SHL+2 site. To maintain the same direction of DNA translocation around the nucleosome but move the dye locations to monitor the movement of DNA into the nucleosome, we made H2A/[end, +12] and H3/[end, +9] constructs where the dye-labeled linker DNA lied "downstream" (to the right) of the 601 sequence (*Figure 5A,B*). The entry-side movement traces measured at 2 μM ATP primarily showed an increase in FRET as expected from the movement of the short linker DNA into the nucleosome, bringing the Cy5 dye closer to Cy3. As before, the entry-side movement traces showed an initial, small apparent decrease in FRET before the major FRET increase, likely due to the photophysical effect on the dye upon enzyme binding (*Figure 3—figure supplement 1*). The FRET increase occurred in a stepwise manner, indicating stepwise DNA translocation into the nucleosome (*Figure 5A,B*). The distributions of the ΔFRET step sizes again showed a major peak around ~0.1 with a tail extending to larger values (*Figure 5C,D*). Fitting to multiple evenly spaced Gaussian peaks as described earlier yielded a step size of ΔFRET = $0.088 \pm 0.002$ with a missed step probability of $f = 22\%$ for the H2A/[end, +12] construct and ΔFRET = $0.084 \pm 0.001$ and $f = 17\%$ for the H3/[end, +9] construct (*Figure 5C,D* and *Figure 5—figure supplement 2A,B*). Assuming that the DNA moved along the canonical nucleosomal wrapping path as it entered the nucleosome, we constructed calibration curves for the two labeling schemes by varying the initial linker DNA lengths and measured slopes of the FRET change versus linker DNA length to be $0.055 \pm 0.002$ bp$^{-1}$ for the H2A/[end, +12] construct and $0.048 \pm 0.004$ bp$^{-1}$ for the H3/[end, +9] construct, which again did not change appreciably in the presence of bound enzyme (*Figure 5—figure supplement 1*). Based on these slopes, the step size distributions were consistent with step sizes of $1.6 \pm 0.1$ and $1.7 \pm 0.2$ bp, respectively, for these constructs.

Finally, to test the dependence of the entry-side step size on the ATP concentration, we monitored remodeling of the H2A/[end,+12] construct at 20 μM and 80 μM ATP, again by increasing the time resolution of our measurements. The entry-side movement traces from these experiments also showed a stepwise increase in FRET (*Figure 5—figure supplement 3*). The resulting step size distributions were consistent with step sizes of $2.0 \pm 0.1$ bp at 20 μM ATP and $1.9 \pm 0.2$ bp at 80 μM ATP (*Figure 5—figure supplement 2A*), again similar to the results measured at 2 μM ATP. Again, the observation that some of the estimated missed step probabilities were larger than would be expected based on our time resolution (*Figure 5—figure supplement 2*) suggests that the DNA may occasionally enter the nucleosome with larger step sizes.

Therefore, our results from experiments using a number of different labeling schemes and ATP concentrations suggested that RSC translocated DNA primarily in ~1–2 bp increments at both the entry side and exit side of the nucleosome.

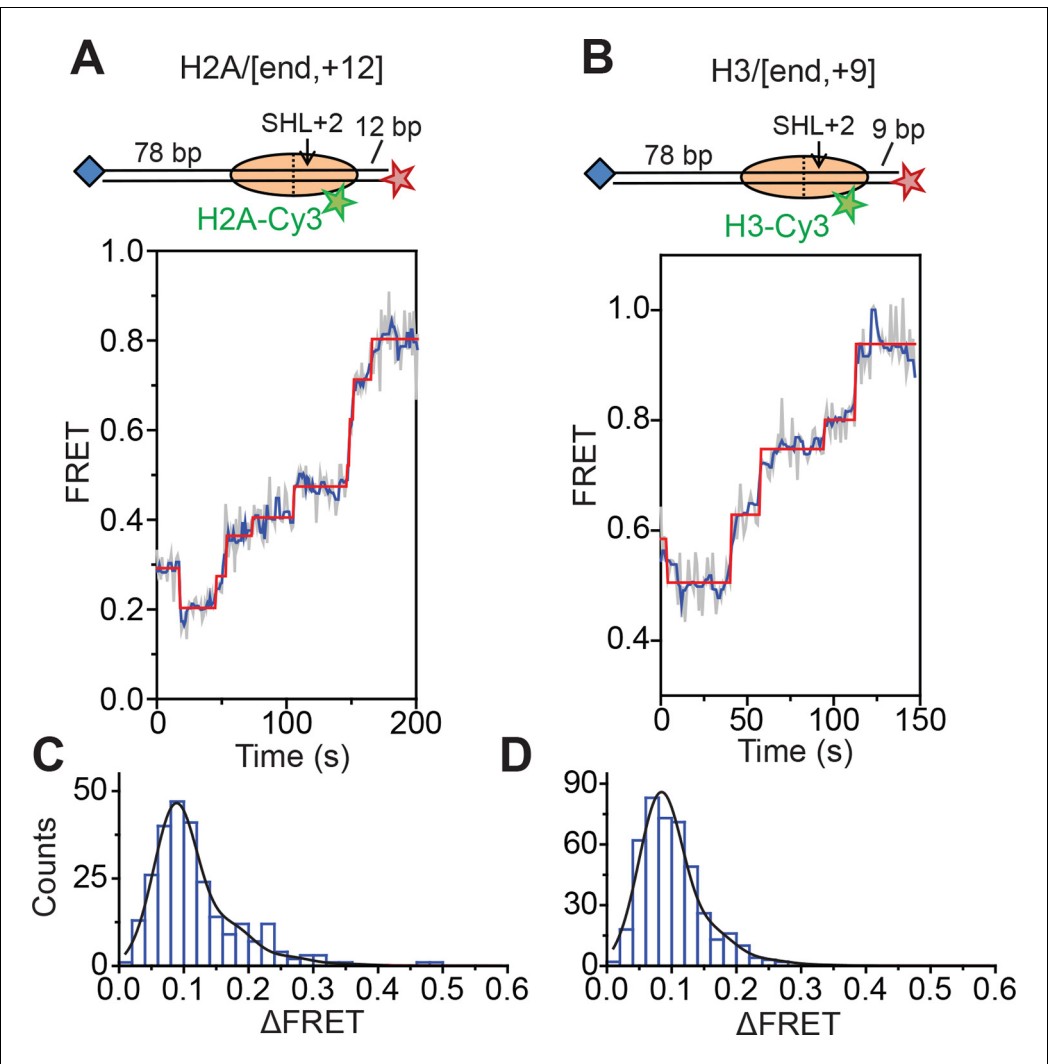

**Figure 5.** DNA enters the nucleosome in a stepwise manner, exhibiting a step size distribution peaked at ~1–2 bp during RSC-mediated remodeling. Remodeling was monitored for H2A/[end,+12] (A and C), and H3/[end,+9] (B and D) constructs. (A and B) Top: diagram of the nucleosome constructs used. Bottom: Representative entry-side movement traces in the presence of 5 nM RSC and 2 µM ATP at 20°C. Light grey, raw FRET data; blue, 5-point median-filtered data; red, fit by the step-finding algorithm. (C and D) Histograms of the measured step sizes in FRET change (blue bars) and the fit to the modeled step size distribution shown in *Equation 1* (black line).

The following figure supplements are available for figure 5:

**Figure supplement 1.** Calibration of FRET values as a function of the linker DNA length for constructs monitoring entry-side movement.

**Figure supplement 2.** Analysis of DNA translocation step sizes of entry-side movement.

**Figure supplement 3.** Monitoring entry-side DNA motion at higher ATP concentrations.

## Discussion

In this study, we used single-molecule FRET to monitor nucleosome remodeling by a prototypical SWI/SNF-family remodeler, RSC. This approach enabled us to track the motion of DNA across individual nucleosomes in real time, providing new insight into the mechanisms by which RSC repositions nucleosomes along DNA.

Our results showed that RSC remodeled mononucleosomes primarily by translocating DNA around the nucleosome under our remodeling conditions (without acceptor nucleosomes, acceptor DNA, transcription factors, chaperones, or other remodeling factors). At the nucleosome edges where DNA enters and exits the nucleosome, RSC did not lift the DNA by a large distance away from the nucleosomal surface, and our data are consistent with translocation of the DNA largely along or close to the canonical nucleosomal wrapping path. In some cases, we observed a small FRET decrease preceding DNA translocation (*Figure 3—figure supplement 1*) that was likely due to changes in the photophysical properties of Cy3 upon enzyme binding. Similarly, constructs designed to report on the dynamics of the H2A-H2B dimer showed no FRET change or a small apparent FRET change likely due to photophysical changes of Cy3 upon enzyme binding. Even if these FRET changes were due to actual distance changes, rather than photophysical changes of the dye, they would still represent minor deviations (~0.5 nm) of DNA from the canonical wrapping path around the octamer or slight repositioning of the H2A-H2B dimer relative to the rest of the octamer. However, because lifting the DNA from the surface of the octamer or displacing the H2A-H2B dimer even by this small distance could disrupt many histone-DNA or histone-histone contacts, we cannot exclude the possibility that the observed translocation of DNA around the nucleosome is associated with the disruption of a substantial number of histone-DNA or histone-histone contacts (*Lorch et al., 2010*). In other words, it is possible that RSC causes DNA unwrapping or lifting of the DNA off the nucleosome, but still holds the unwrapped/lifted DNA near the surface of the nucleosome. However, such a small-distance motion would have only a minor effect on our measured FRET signal. These results are also consistent with an electron microscopy structure of the RSC complex, which suggests a tight fit of the nucleosome in the central cavity of the remodeler that would not seem to accommodate large-scale displacement of DNA from the nucleosome surface (*Leschziner et al., 2007*).

Thus, we characterized the step size of DNA translocation into and out of the nucleosome by assuming that the DNA moved along the canonical path around the nucleosome during RSC-mediated remodeling. We observed a step size distribution that peaked at a step size of ~1–2 bp both for entry-side and exit-side movement though movements with larger step sizes were also observed. Even if the motion of the DNA is not exactly along the canonical path, the small ΔFRET step sizes of 0.08–0.11 that we observed in the single-nucleosome remodeling traces likely correspond to a distance change of ~0.3–0.6 nm (assuming a previously measured Förster radius of 6 nm for the Cy3-Cy5 FRET pair (*Murphy et al., 2004*)). Based on the geometry of our labeling schemes and the crystal structure of nucleosomes containing the 601 positioning sequence (*Chua et al., 2012*), translocation by 1 bp should be associated with changes to the Cy3-Cy5 distance of ~0.3 nm on average, so our data are again consistent with a DNA translocation step size of ~1–2 bp. Although changes to the photophysical properties of the dyes or the relative orientation of the dyes could potentially affect our interpretation of the FRET changes, many of these effects would be expected to depend on the specific labeling scheme. The fact that we observed similar step-size histograms with several different attachment sites of the Cy5 dye on the DNA and different attachment sites of the Cy3 dye on the octamer suggests that the measured step sizes likely reflect the true translocation step sizes of DNA around the nucleosome during RSC-catalyzed remodeling. It has been shown previously that some SF2 ATPase-containing DNA translocases or helicases translocate DNA in 1 bp increments (*Myong et al., 2007*; *Gu and Rice, 2010*; *Rajagopal et al., 2010*; *Cheng et al., 2011*; *Deindl et al., 2013*), so the SF2 ATPase domain of RSC might also translocate DNA at the SHL2 site of the nucleosome in 1 bp steps. The reason that the step sizes that we observed for RSC-mediated nucleosome translocation at the nucleosomal edges were not exactly 1 bp could potentially be due to the following reasons: 1) missing steps due to our limited time resolution could cause two consecutive 1 bp steps to appear as a 2 bp step, 2) two 1 bp DNA distortions generated at the SHL2 site could merge together while transiting to the entry/exit site, and 3) the DNA translocation path could deviate slightly from the canonical wrapping path, making our calibration, which was determined based on canonically wrapped nucleosomes, slightly off.

It has been shown previously that RSC translocates DNA at the SHL2 site where the enzyme's ATPase domain engages the nucleosome (*Saha et al., 2002*; *Saha et al., 2005*; *Zofall et al., 2006*; *Dechassa et al., 2008*) and that the isolated RSC catalytic subunit translocates naked DNA in ~2 bp increments (*Sirinakis et al., 2011*). Based on these previous findings, it is reasonable to expect that the RSC complex may translocate DNA around the nucleosome in 1–2 bp steps, as has been proposed before (*Saha et al., 2005*). However, such small steps had not been previously observed

during nucleosome remodeling by SWI/SNF enzymes, so it was unclear whether the small DNA translocation step size by the ATPase at the SHL2 site would result in small DNA movement steps around the nucleosome, especially given that the entry and exit sites of nucleosome reside ~50 and 100 bp away from the SHL2 site. Indeed, in the context of remodeling intact nucleosomes, a previous biochemical study showed that SWI/SNF complexes translocate DNA around the nucleosome in large, ~50 bp steps (*Zofall et al., 2006*). Because our current observations suggest that RSC translocates DNA around the nucleosome primarily in steps of 1–2 bp, the previously observed ~50 bp translocation steps likely do not reflect the fundamental step size of nucleosome translocation by RSC. Rather, they may be compound steps composed of many ~1–2 bp steps and may reflect relatively long kinetic pauses of DNA translocation imposed by the energy landscape of the nucleosomal substrates. However, we note that because the cavities in the RSC and SWI/SNF complexes that accommodate nucleosomes appear somewhat different (*Leschziner et al., 2007*; *Chaban et al., 2008*; *Dechassa et al., 2008*), it is possible that SWI/SNF and RSC do not translocate DNA around the nucleosome in exactly the same manner. Furthermore, it is also worth noting that our measurements were performed at limiting ATP concentrations, up to roughly the $K_m$ value for ATP (~80 µM) of RSC (*Cairns et al., 1996*), and it is possible that at higher ATP concentrations, other processes could become rate limiting and alter the behavior of the complex. Nevertheless, the behavior at limiting ATP concentrations likely reflects the response of the nucleosome to the individual translocation events catalyzed by the ATPase at the SHL2 site.

Because the SF2 ATPase of RSC likely translocates DNA 1 bp at a time at the SHL2 site of the nucleosome, our observation that RSC moves the DNA into and out of the nucleosome primarily in ~1–2 bp increments suggests that DNA motion across the nucleosome from the nucleosomal entry site to the nucleosomal exit site is likely coupled directly to DNA translocation at the SHL2 site by the ATPase domain. Because we tracked the motion of the DNA only at the nucleosomal edges, our data cannot determine whether an intranucleosomal loop or bulge was formed inside the nucleosome. If such loops exist, our observations of the small step sizes of DNA translocation at both entry and exit sites of the nucleosome place constraints on such internal DNA loops/bulges: these loops/bulges are either small and most often contain 1–2 bp of extra DNA; or if they are not so small, they must accumulate and release gradually, primarily 1–2 bp at a time at the nucleosome entry and exit sites, which would require a specific mechanism to extrude a large internal loop to the exit side in small steps.

Finally, we note that although the nucleosome translocation step size observed here for RSC is similar to the nucleosome translocation step size previously observed for ISWI-family enzymes (*Deindl et al., 2013*), these results do not necessarily suggest that the remodeling mechanisms by these two families of chromatin remodelers are entirely similar. For both enzyme families, the observed nucleosome translocation step sizes likely resulted from the intrinsic DNA translocation step size of their ATPase domains, which both belong to the SF2 helicase family. However, it has been shown previously that the SWI/SNF-family remodelers disrupt the DNA-octamer contacts to a much greater extent than the ISWI-family remodelers (*Fan et al., 2003*; *Dechassa et al., 2012*). Our results suggest that, despite such disruption, the RSC enzyme likely still holds the DNA close to the nucleosome and hence, such disruption does not strongly perturb the step sizes with which DNA is translocated across the nucleosome.

## Materials and methods

### Histone purification and labeling

Recombinant *X. laevis* histones were expressed in BL21 (DE3) pLysS cells (Promega, Madison, WI) and purified under denaturing conditions. Briefly, we isolated inclusion bodies and extracted the histones as described previously (*Luger et al., 1999*), then dialyzed the histones into buffer A (7 M urea, 20 mM Na-HEPES pH 8.0, 100 mM NaCl, 1 mM DTT and 1 mM Na-EDTA). This solution was then loaded onto HiTrap-Q cation exchange and ResourceS anion exchange columns (GE Healthcare, Pittsburgh PA) connected in series (*Wittmeyer et al., 2004*). After washing with buffer A, the HiTrap-Q column was then removed before the protein was eluted by gradually increasing the concentration of NaCl. To generate histones site-specifically labeled with Cy3, plasmids for the expression of H2A K119C, H2B T49C, and double mutant H3 (G33C+C110A) were created by site-directed

mutagenesis. These constructs were purified and labeled with sulfo-Cy3 maleimide or sulfo-Cy5 mal-eimide (GE Healthcare) under denaturing conditions (*Hwang et al., 2014*). The histone H2A and H3 mutants have been previously used for FRET-based studies of nucleosome remodeling (*Yang et al., 2006*; *Rowe and Narlikar, 2010*). Histone octamer was reconstituted with an ~1:1 ratio of labeled and unlabeled histone in order to maximize the yield of singly-labeled octamer and purified by gel filtration chromatography as described previously (*Luger et al., 1999*; *Hwang et al., 2014*). For the reconstitution of the H2A/H2A and H2B/H2B constructs, histone octamer was reconstituted with an ~1:1 ratio of Cy3-labeled and Cy5-labeled histone.

## Preparation of DNA constructs

DNA constructs were made by PCR from a plasmid containing a modified 601 positioning sequence (*Lowary and Widom, 1998*; *Partensky and Narlikar, 2009*). The PCR primers contained 5' Cy5 or biotin-TEG modifications (IDT, Coralville, IA) to install these modifications at the indicated locations. For constructs containing gaps or backbone Cy5 labels, the DNA was constructed by annealing a set of overlapping oligonucleotides (IDT) and ligating them into a single double-stranded DNA construct (*Hwang et al., 2014*). Backbone Cy5 labels were inserted opposite guanosine residues. DNAs were purified by PAGE.

## Preparation of nucleosome constructs

Mononucleosomes were assembled by the salt dialysis method after mixing labeled octamer and DNA at a 1.2:1 ratio and purified by glycerol gradient centrifugation as previously described (*Lee and Narlikar, 2001*).

## RSC purification

RSC was purified from *S. cerevisiae* strain BCY211 that expresses a TAP-tagged Rsc2 subunit follow-ing published protocols (*Saha et al., 2002*; *Wittmeyer et al., 2004*). Protein concentration was quantified by Sypro Red staining and comparison with BSA standards.

## Single-molecule FRET measurements

Quartz slides were cleaned, functionalized with a 1:100 mixture of biotin-PEG:PEG (Laysan Bio, Arab, AL), and assembled into flow chambers as previously described (*Joo and Ha, 2008*). Nucleo-somes were immobilized on the biotin-PEG on the slide via streptavidin. The sample was excited with a 532 nm Nd:YAG laser (CrystalLaser, Reno, NV) on a custom built total internal reflection fluo-rescence (TIRF) microscope (*Deindl and Zhuang, 2012*). Fluorescence emission was collected with a 60x water immersion objective (Olympus, Tokyo, Japan), filtered with a 550 nm long-pass filter (Chroma Technology, Bellows Falls, VT), split with a 630 nm dichroic mirror (Chroma Technology), and imaged onto two halves of an Andor iXon+888 EM-CCD camera (Andor Technology, Belfast, UK). Imaging was done at 30°C (unless otherwise specified) in imaging buffer (40 mM Tris,12 mM HEPES, pH 7.5, 60 mM KCl, 3 mM MgCl2, 10% (v/v) glycerol, 0.02% (v/v) Igepal CA-630, 10% (w/v) glucose, 2 mM trolox, 0.1 mg/mL acetylated BSA), supplemented with an oxygen scavenging system (800 µg/mL glucose oxidase, 50 µg/mL catalase) (*Rasnik et al., 2006*; *Blosser et al., 2009*). Images were collected at 1 Hz, except for the data in *Figure 4—figure supplement 4* and *Figure 5—figure supplement 3* which were taken at 8 Hz (for the experiments with 20 µM ATP) or 16 Hz (for the experiments with 80 µM ATP). The field of view was illuminated with ~25–50 W/cm$^2$ of laser light for data collected at 1 Hz. For experiments done at 8 Hz and 16 Hz, ~400 and 800 W/cm$^2$, respectively, was used.

  In experiments where the FRET acceptor dye is on the DNA and the FRET donor is on the histone (H2A, H2B or H3), we observed three populations of labeled nucleosomes: nucleosomes with a sin-gle donor dye residing on the H2A, H2B or H3 subunit proximal to the short linker, nucleosomes with a single donor dye residing on the H2A, H2B or H3 subunit distal to the short linker, and nucle-osomes with two donor dye molecules (*Blosser et al., 2009*). These nucleosomes can be distin-guished on the basis of their FRET value, and we select only those nucleosomes with the Cy3 on the proximal H2A, H2B or H3 subunit for analysis, which gave the highest FRET values. For the H2A/H2A and H2B/H2B constructs, we selected those showing FRET > 0.2 and FRET > 0.1, respectively, as

nucleosomes lacking Cy5 show zero FRET and nucleosomes lacking Cy3 are not visible during 532 nm illumination.

## Data analysis

Single-molecule FRET traces were generated and analyzed with IDL (ITT Visual Information Solutions, Boulder, CO) as described previously (*Deindl and Zhuang, 2012*) (code available at http://zhuang. harvard.edu/smFRET.html). Single nucleosomes were identified by selecting traces that showed one-step photobleaching. The fluorescence intensity after photobleaching was used for background subtraction. To classify traces as entry-side or exit-side movement traces, we manually identified traces exhibiting remodeling, smoothed the data with a 3 pt median filter, then classified them in an automated manner using the following criteria: If a trace showed 3 consecutive points that were greater than the initial FRET by 0.1 (0.07 for the H2A/[backbone,–15] data) before reaching the final FRET value, indicating an increase in FRET followed by FRET decrease, it was classified as an entry-side movement trace for the H2A/[end,+6] and H3/[end,+6] constructs or an exit-side movement trace for the H2A/[backbone,–15] construct. If the trace reached the final FRET value without such an initial increase, indicating a monotonic FRET decrease, the trace was classified as an exit-side movement traces for the H2A/[end,+6] and H3/[end,+6] constructs or an entry-side movement trace for the H2A/[backbone,–15] construct.

In the experiments to determine the step size of remodeling, pauses in the FRET traces were identified using a previously developed step-finding algorithm based on Chi-square minimization (*Kerssemakers et al., 2006*). We also identified steps using an alternative, hidden Markov model-based method (*McKinney et al., 2006*). Only the steps identified by these algorithms that satisfy the following criteria were included for further analysis: 1) the pauses are not shorter than 5 frames, and 2) the pauses occurred at FRET values that are inside the range defined by the calibration curves in *Figure 4—figure supplement 2* and *Figure 5—figure supplement 1* where FRET varies linearly with distance from the nucleosome. Occasionally, the step finding algorithm identified steps in the backward direction. Because these events were rare (accounting for less than 5% of steps), we did not include these steps in our analysis.

To simulate the step size distribution resulting from a uniform step size and a fixed probability of missing short pauses, causing larger steps whose size is an integer multiple of the step size, the observed step size histograms were fit to the function:

$$y = \sum_{n=1}^{6} A\, f^{n-1} \exp\left(-\frac{(x - n\,c)^2}{2\,s^2}\right) \tag{1}$$

where *f* is the probability that a pause is missed, *c* is the step size, and *s* is the standard deviation of the step size. The *s* parameter was fit globally across all step size histograms showing the same type of motion (i.e. exit-side movement or entry-side movement).

## Ensemble FRET measurements

Ensemble FRET measurements were performed by monitoring the Cy5 intensity at 670 nm under 532 nm excitation in a Cary Eclipse Fluorescence Spectrophotometer (Varian, Palo Alto, CA). Reactions were performed in imaging buffer, and initiated by the addition of RSC and ATP to a solution containing nucleosomes. Data were normalized by scaling the initial Cy5 intensity to 1.0 and the final steady state Cy5 intensity to 0.

## Acknowledgements

We thank Geeta Narlikar for providing the yeast strain for RSC expression and Benjamin Altheimer for a critical reading of the manuscript. This work was supported in part by the National Institutes of Health (GM105637 to XZ and GM 48413 to BB). WLH acknowledges support from the NIH T32G007753 Training Grant. SD was a Merck Fellow of the Jane Coffin Childs Foundation. XZ is a Howard Hughes Medical Investigator.

# Additional information

## Competing interests

XZ: Reviewing editor, *eLife.* The other authors declare that no competing interests exist.

## Funding

| Funder | Grant reference number | Author |
| --- | --- | --- |
| National Institute of General Medical Sciences | GM105637 | Xiaowei Zhuang |
| Howard Hughes Medical Institute | | Xiaowei Zhuang |
| National Institute of General Medical Sciences | GM 48413 | Blaine Bartholomew |

The funders had no role in study design, data collection and interpretation, or the decision to submit the work for publication.

## Author contributions

BTH, Designed the research, Performed the research, Wrote the paper, Conception and design, Acquisition of data, Analysis and interpretation of data, Drafting or revising the article; WLH, Helped with acquisition of data; Helped to perform the research; SD, BB, Helped with design of the experiments; NC, Helped with enzyme purification, Aided with RSC purification; XZ, Designed the research, Wrote the paper, Conception and design, Analysis and interpretation of data, Drafting or revising the article

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
