## [Decision Letter]

Thank you for submitting your work entitled "Stepwise nucleosome translocation by RSC remodeling complexes" for peer review at *eLife*. Your submission has been favorably evaluated by Jim Kadonaga (Senior editor), a Reviewing editor, and three reviewers.

The reviewers have discussed the reviews with one another and the Reviewing editor has drafted this decision to help you prepare a revised submission.

In this manuscript, the Zhuang and Bartholomew groups employ a single molecule FRET method to probe remodeling by RSC. Notably, these same groups have previously used this same method to probe remodeling by ISWI-like enzymes. The authors report that RSC, like ISWI, appears to translocate DNA in 1-2 bp steps from both the entry and exit DNA positions. By using different locations for the FRET probes, the authors present evidence that DNA translocation does not grossly displace DNA from the H2A C-terminal tail or the H3-N-terminal tail suggesting that DNA is not displaced at the entry/exit positions of the nucleosome. The authors also do not see loss of the H2A-C-terminal tail from DNA during remodeling, suggesting that the H2A/H2B dimer is not displaced.

Overall, the approaches are very well designed and executed and of high technical merit. The choices of label locations etc. and the use of gapped substrates generate a system that provides clear and classifiable outputs. Furthermore, the paper is written clearly, and the figures are well presented. Overall, this is a very well executed, interesting, and satisfying paper. The work provides the clearest and most direct evidence to date for the step size and DNA path on nucleosomes; the use of 1-2 bp steps, and that this 1-2 bp DNA wave/loop likely follows a generally canonical path around the nucleosome. The work helps take the field forward in a clear manner.

In general, the experiments are elegantly designed, with numerous controls to back up many of the authors' conclusions. And although the manuscript is generally well written, a few areas could be clarified and/or expanded further:

1) While the model proposed by the authors is plausible and consistent with the data, the argument that the FRET data rule out a mechanism where DNA is peeled off the mononucleosome was not totally convincing. This is a main criticism of the work. Since this argument is central to the conclusions of the manuscript (it also affects interpretation of the stepping data), the authors should devote more text to strengthening it. In this regard, one surprising result presented here is that RSC remodeling appears to be identical to ISWI, using this particular FRET assay. There are a host of studies demonstrating that RSC-like enzymes are more disruptive to nucleosomes than ISWI. Thus, it seems that this assay is unable to detect key aspects of the remodeling reaction. Notably, the authors' assay only monitors DNA translocation at the extreme nucleosome edge. What happens elsewhere on the nucleosome is invisible. For instance, it remains a possibility that RSC and ISWI both draw in DNA at the nucleosome edge (1-2 bp per steps), but for RSC this DNA would then be extruded into a large, constrained loop on the nucleosome surface. Such a dramatic structure would be undetectable in this assay. It was surprising that the authors had no discussion of the similarities between ISWI and RSC using this assay – perhaps because the results are inconsistent with so many other studies. Perhaps the authors can investigate loops by looking at the timing of exit/entry DNA translocation as they did for ISWI-like enzymes previously.

Furthermore, related to the above paragraph, the authors make a very strong conclusion that RSC action does not displace DNA from the octamer surface and that there is no substantial displacement of the dimers. These are key conclusions, as they are the only ones that really speak to the novelty of this work (other studies have already dealt with step size). These conclusions are much too strong and not supported well by data. The authors do not actually probe the octamer surface. All histone FRET probes are present on either the start of the H3 N-terminal domain (residue 33), or the H2A C-terminal domain (residue 119). Both residues lie within a basic patch of amino acids that make contact with entry/exit DNA. Thus, the experiments only show that the H3 or H2A tails do not dissociate from entry/exit DNA. It is possible that these regions may remain associated with DNA even if it is displaced from the octamer, and the authors may be missing this event. FRET probes should be placed on solvent expose octamer surfaces that are not part of flexible domains.

2) For studies on step-size, the authors used extremely low ATP concentrations (<10x below Km) in order to slow down the reaction. Previously studies from the Narlikar lab, as well as in the previous single molecule studies from Bustamonte, showed that ATP concentration as a large impact on pausing events between bursts of translocation. The worry is that very low ATP concentrations may change the rate-limiting step of the reaction, leading to aberrant step sizes that are normally not a part of the reaction. The authors should investigate the impact of ATP concentration on their results. The authors should also comment on why some experiments utilize 5 micromolar ATP (single traces), but others use 20 micromolar (ensemble reactions or pooling of traces).

3) Prior papers support the model proposed in this paper: the use of ~1-2bp step size via a coupled mechanism, from a directional DNA translocase operating from SHL2. For example, one arrives at this model for RSC by simply combining Saha et al., NSMB 2005 (directional DNA translocation from SHL2, with gaps used in multiple ways as evidence) with Sirinakis et al., 2011 (providing the 1-2bp translocation step size on DNA via single molecule approaches). Both of those papers interpret and favor that model, and that model has been featured in major reviews. The authors might more clearly acknowledge in the Discussion the existing data already underlying this model, and then how their work extends.

4) A prior EM structure of RSC complex by Leschziner and colleagues is highly supportive of the results in this paper and the model. In this structure, a large pocket resides within RSC complex, of nearly perfect dimensions of a nucleosome, and there is just enough room around the nucleosome in that pocket to also accommodate the propagation of a small DNA loop/wave of a few base pairs around the nucleosome, but not larger DNA wave/loop sizes, or the large unwrapping of DNA at the entry site. This existing structural support for the results and model would be worth mentioning in the Discussion.

Specific editorial comments/requests (listed in order of appearance):

1) The authors observe two types of FRET traces, one with a decrease in FRET vs. one with an increase followed by a decrease, which they attribute to the two possible directions of RSC translocation, termed "entry-" and "exit-side movement". This interpretation is corroborated by control experiments in which a 2-nt gap has been introduced in each DNA strand, blocking each one direction of translocation. In the schematic describing these measurements (Figure 1, Figure 1—figure supplement 3), the gaps in DNA are located on opposite strands. Do SWI/SNF remodelers track a particular DNA strand? This is not discussed in the text and should be clarified.

2) In Figure 1 (and elsewhere) the time trace shows the Cy3 fluorescence signal extinguish at ~80 s. Is the loss of signal due to dissociation of the nucleosome from DNA or is this photobleaching? Is there any evidence that nucleosomes dissociate from the DNA due to RSC and if so, is this observed for both entry- and exit-side movements? This is not discussed adequately.

3) In the text (Results) the authors state that the "[2-nt] gap at SHL+2, eliminated the zero FRET population and resulted in a single population with a FRET of ~0.14 after remodeling". However, there appears to be a small peak at 0 FRET in Figure 1. What is this due to? According to the text, the SHL+2 gap site should favor "entry-side" motion, which should not generate a zero-FRET state.

4) In Figure 2, the histogram for FRET labeled nucleosomes in the presence of RSC, but no ATP (in red) seems to be bimodal, with a FRET state at ~0.3 and ~0.55, but this is not discussed in the text. Is this again attributable to photophysical effects, as rationalized for the small decrease in FRET with the addition of RSC? Or is this evidence for multiple conformations?

5) In the text (Results) the authors claim that "If RSC were to unpeel a substantial amount of DNA and lift the DNA off the nucleosomal surface by a large distance, we would expect the FRET traces to exhibit a FRET decrease as the Cy5 dye on the linker DNA moves away from the nucleosome where the Cy3 dye resides". This statement is not quite as obvious as stated. This claim seems highly dependent on the geometry of the DNA-nucleosome complex (e.g. the locations of the nucleosome and DNA fluorescence labels) and seems to assume that the unpeeled DNA is fully extended. One could imagine situations where the unpeeled DNA contributes very little to the FRET or could even increase FRET, especially if the DNA end can bend (or loop) back on itself. This will obviously depend on the length / flexibility of DNA. As this is an important point of the manuscript that affects subsequent data interpretation, the authors should devote more effort to clarifying it and supporting it with more quantitative arguments.

On a related note, the authors repeatedly refer to "small-" and "large-" distance lifting of DNA of the nucleosome surface, but some numbers should be provided here. Although they provide an estimated bound of 0.5 nm from FRET, this seems to refer to inter-dye distance, which is not necessarily the same as the contour length of DNA lifted (especially if bending/looping is considered).

6) The authors used a step-finding algorithm to determine the translocation step size of RSC. Were any selections made for the steps in their analysis or were all steps detected (regardless of size) shown in the histograms (Figure 4 and Figure 5)? There is a concern that such an algorithm could biasthe step size distribution, by not reliably detecting smaller step sizes for instance. Did the authors test out other algorithms (e.g. t-test based step-finding, etc.) or try a pairwise distribution? If so, was the step size robust against detection method?

---

## [Author Response]

*[…] In general, the experiments are elegantly designed, with numerous controls to back up many of the authors' conclusions. And although the manuscript is generally well written, a few areas could be clarified and/or expanded further:*

*1) While the model proposed by the authors is plausible and consistent with the data, the argument that the FRET data rule out a mechanism where DNA is peeled off the mononucleosome was not totally convincing. This is a main criticism of the work. Since this argument is central to the conclusions of the manuscript (it also affects interpretation of the stepping data), the authors should devote more text to strengthening it. In this regard, one surprising result presented here is that RSC remodeling appears to be identical to ISWI, using this particular FRET assay. There are a host of studies demonstrating that RSC-like enzymes are more disruptive to nucleosomes than ISWI. Thus, it seems that this assay is unable to detect key aspects of the remodeling reaction. Notably, the authors' assay only monitors DNA translocation at the extreme nucleosome edge. What happens elsewhere on the nucleosome is invisible. For instance, it remains a possibility that RSC and ISWI both draw in DNA at the nucleosome edge (1-2 bp per steps), but for RSC this DNA would then be extruded into a large, constrained loop on the nucleosome surface. Such a dramatic structure would be undetectable in this assay. It was surprising that the authors had no discussion of the similarities between ISWI and RSC using this assay – perhaps because the results are inconsistent with so many other studies. Perhaps the authors can investigate loops by looking at the timing of exit/entry DNA translocation as they did for ISWI-like enzymes previously.*

We thank the reviewers for pointing this out. In fact, we did not mean to imply that our data rule out the unpeeling of DNA from the mononucleosome. Based on our data, we suggested that the DNA is not lifted away from the surface by a large distance. We actually stated explicitly in our manuscript that our data do not exclude the possibility that the enzyme disrupts many of the histone-DNA contacts simultaneously during remodeling, but still holds the unpeeled DNA near the surface of the nucleosome. We have now made this point clearer both in the Results and Discussion sections with the following statements:

Results section: “Although our data are not consistent with RSC moving the DNA at the nucleosome edges by a large distance away from the octamer surface, as would be expected if RSC were to unwrap a substantial amount of DNA from the edge of the nucleosome and allow the DNA to follow its unconstrained path, our data do not exclude the possibility that the enzyme disrupts many of the histone-DNA contacts simultaneously during remodeling (Lorch, et al., 2010), but still holds the unwrapped or lifted DNA near the surface of the nucleosome.”

Discussion section: “However, because lifting the DNA from the surface of the octamer or displacing the H2A-H2B dimer even by this small distance could disrupt many histone-DNA or histone-histone contacts, we cannot exclude the possibility that the observed translocation of DNA around the nucleosome is associated with the disruption of a substantial number of histone-DNA or histone-histone contacts (Lorch, et al., 2010). In other words, it is possible that RSC causes DNA unwrapping or lifting of the DNA off the nucleosome, but still holds the unwrapped/lifted DNA near the surface of the nucleosome. However, such a small-distance motion would have only a minor effect on our measured FRET signal.”

Although our data showed that the step size with which RSC translocates DNA around the nucleosome is similar to that of the ISWI enzymes, we do not imply that the remodeling mechanisms used by these two families of enzymes are identical. In light of the reviewers’ comment, we also added an explicit discussion about this point as the last paragraph of the Discussion. It is stated as below:

“Finally, we note that although the nucleosome translocation step size observed here for RSC is similar to the nucleosome translocation step size previously observed for ISWI-family enzymes (Deindl, et al., 2013), these results do not necessarily suggest that the remodeling mechanisms by these two families of chromatin remodelers are entirely similar. […] Our results suggest that, despite such disruption, the RSC enzyme likely still holds the DNA close to the nucleosome and hence, such disruption does not strongly perturb the step sizes with which DNA is translocated across the nucleosome.”

We also agree with the reviewers’ comment that we only probed the DNA movement at the edges of the nucleosome where DNA enters and exits the nucleosome, and our data does not rule out the possibility of the formation of intranucleosomal loops or bulges. We had already acknowledged this possibility but stated that if such intranuclesomal loops formed, they should accumulate and release gradually because we observed 1-2 bp translocation steps at both entry and exit sides of the nucleosome. We have now further expanded this part of the Discussion as below:

“Because we tracked the motion of the DNA only at the nucleosomal edges, our data cannot determine whether an intranucleosomal loop or bulge was formed inside the nucleosome. If such loops exist, our observations of the small step sizes of DNA translocation at both entry and exit sites of the nucleosome place constraints on such internal DNA loops/bulges: these loops/bulges are either small and most often contain 1-2 bp of extra DNA; or if they are not so small, they must accumulate and release gradually, primarily 1-2 bp at a time at the nucleosome entry and exit sites, which would require a specific mechanism to extrude a large internal loop to the exit side in small steps.”

Finally, as we discussed in the manuscript (Discussion, second paragraph), our FRET data are sufficient to rule out larger distance changes to the path of the DNA that would affect our interpretation of the step sizes. Even if the movement of the DNA deviated from its canonical path, the small FRET changes that we observed in each step (~0.08 – 0.11) are consistent with the distance changes (~0.3 – 0.6 nm) expected for the translocation of 1-2 bp of DNA, as we discussed in the third paragraph of the Discussion.

*Furthermore, related to the above paragraph, the authors make a very strong conclusion that RSC action does not displace DNA from the octamer surface and that there is no substantial displacement of the dimers. These are key conclusions, as they are the only ones that really speak to the novelty of this work (other studies have already dealt with step size). These conclusions are much too strong and not supported well by data. The authors do not actually probe the octamer surface. All histone FRET probes are present on either the start of the H3 N-terminal domain (residue 33), or the H2A C-terminal domain (residue 119). Both residues lie within a basic patch of amino acids that make contact with entry/exit DNA. Thus, the experiments only show that the H3 or H2A tails do not dissociate from entry/exit DNA. It is possible that these regions may remain associated with DNA even if it is displaced from the octamer, and the authors may be missing this event. FRET probes should be placed on solvent expose octamer surfaces that are not part of flexible domains.*

We thank the reviewers for this suggestion. Following this suggestion, we generated a construct where the dye label is moved from the histone tail to within the globular domain of histone H2B at position 49. We made two nucleosome constructs with this new histone labeling scheme: 1) The H2B/H2B construct where one of the H2B is labeled with the FRET donor Cy3 and the other is labeled with the FRET acceptor Cy5. Upon addition of RSC and ATP, we did not observe any FRET change, suggesting that the H2A-H2B dimer is not substantially displaced from the nucleosome during remodeling, consistent with the results we obtained for the H2A/H2A construct. 2) The H2B/[end, +0] construct, containing Cy3 on the H2B subunit and a Cy5 at the 5’ end of the DNA. Upon addition of RSC and ATP, this new construct again showed entry-side and exit-side movement traces similar to those observed from the H2A/[end, +6] construct (with the Cy3 on the H2A subunit and the Cy5 near the 5’ end of the DNA) and the H3/[end, +6] construct (with the Cy3 on the H3 subunit and the Cy5 near the 5’ end of the DNA). These results suggest that our results obtained from the H2A/[end, +6] and H3/[end, +6] constructs are not due to the histone tails tracking the DNA and masking the remodeling-induced movement. These new results are described in the third paragraph of the subsection “Probing for potential H2A-H2B dimer movement during remodeling”, as well as in Figure 2—figure supplement 2.

Again, we note that we did not conclude in this work that RSC does not displace DNA from the octamer surface, but only that DNA is not displaced from the octamer surface by a large distance. Please see our response to the first part of comment 1 above.

Also we note that our result that RSC translocates DNA around the nucleosome in 1-2 bp steps is novel. Previously, a ~2 bp translocation step size has been observed for the isolated catalytic subunit of RSC translocating on naked DNA without octamer (Sirinakis, et al., 2011), but such small step sizes have not been previously observed in the context of full SWI/SNF-remodeling complexes acting on intact nucleosomes. Indeed, before our study, it was unclear whether the RSC complex would translocate DNA across the nucleosome with a similar step size. In fact, a previous biochemical study showed that a similar SWI/SNF complex translocates DNA around the nucleosome in large, ~50 bp steps (Zofall, et al., 2006). Our current observations suggest that RSC translocates DNA around the nucleosome in 1-2 bp steps, at both nucleosomes edges where DNA enters and exits the nucleosome. Thus, our results suggests that the previously observed, large ~50 bp translocation steps likely do not reflect the fundamental step size of nucleosome translocation by RSC. Rather, they may be compound steps composed of many 1-2 bp steps and reflect relatively long kinetic pauses of DNA translocation imposed by the energy landscape of the nucleosomal substrates.

*2) For studies on step-size, the authors used extremely low ATP concentrations (<10x below Km) in order to slow down the reaction. Previously studies from the Narlikar lab, as well as in the previous single molecule studies from Bustamonte, showed that ATP concentration as a large impact on pausing events between bursts of translocation. The worry is that very low ATP concentrations may change the rate-limiting step of the reaction, leading to aberrant step sizes that are normally not a part of the reaction. The authors should investigate the impact of ATP concentration on their results. The authors should also comment on why some experiments utilize 5 micromolar ATP (single traces), but others use 20 micromolar (ensemble reactions or pooling of traces).*

Following the reviewers’ suggestions, we have determined the step size of the remodeler at two additional concentrations of ATP, 20 µM and 80 µM. The K_m_ of RSC for ATP was previously reported to be 77µM (Cairns et al., 1996), and thus the highest ATP concentration we used here, 80 µM, is comparable the K_m_ value. For both exit-side movement traces and entry-side movement traces, we observed similar step sizes (1-2 bp) at 20 µM and 80 µM ATP to those we observed at 2 µM ATP. The results of these new experiments are reported in Figure 4—figure supplement 3,Figure 4—figure supplement 4 and Figure 5—figure supplement 2,Figure 5—figure supplement 3, as well as in the fourth and seventh paragraphs of the subsection “Characterization of the step size of DNA translocation”. It is difficult for us to extend the measurements to even higher ATP concentrations because the translocation pauses would become too short for us to unambiguously detect with our limited time resolution.

We also added a note in the Discussion section (fourth paragraph) to state that because our experiments are performed at limiting ATP concentrations, up to roughly the K_m_ value of ATP for RSC, it is thus possible that at higher ATP concentrations, other processes could become rate limiting and alter the behavior of the complex. Nevertheless, the behavior at limiting ATP concentrations likely reflects the response of the nucleosome to the individual translocation events catalyzed by the ATPase at the SHL2 site.

With regards to the use of 5 µM ATP and 20 µM ATP in different experiments, we note that the results obtained at 20 µM ATP can be reproduced at 5 µM ATP. Please see Figure 6. Many of the statistical analyses are shown with data taken using 20 µM ATP because it is more efficient for us to accumulate statistics at this condition.

Author response image 1. Comparison of remodeling at 5 and 20 µM ATP.(**A**) Histogram showing the distribution of FRET values before (blue) and after (red) addition of 1 nM RSC and 5 µM ATP (top) or 1 nM RSC and 20 µM ATP (bottom). The data in the bottom panel are reproduced from Figure 1 for comparison. (**B**) The fraction of exit-side movement (cyan) and entry-side movement (orange) traces observed with 1 nM RSC and the indicated concentration of ATP on H2A/[end, +6] nucleosomes. Error bars represent the standard error from >100 nucleosomes per construct. The data for the 20 µM ATP condition are reproduced from Figure 1 for comparison.**DOI:**
http://dx.doi.org/10.7554/eLife.10051.022

*3) Prior papers support the model proposed in this paper: the use of ~1-2bp step size via a coupled mechanism, from a directional DNA translocase operating from SHL2. For example, one arrives at this model for RSC by simply combining Saha et al., NSMB 2005 (directional DNA translocation from SHL2, with gaps used in multiple ways as evidence) with Sirinakis et al., 2011 (providing the 1-2bp translocation step size on DNA via single molecule approaches). Both of those papers interpret and favor that model, and that model has been featured in major reviews. The authors might more clearly acknowledge in the Discussion the existing data already underlying this model, and then how their work extends.*

We have discussed the Sirinakis et al., 2011 paper in our original submission. As suggested by the reviewers here, we have further elaborated this discussion and combined with the discussion on the Saha et al., NSMB 2005 paper. In the Discussion section, we stated:

“It has been shown previously that RSC translocates DNA at the SHL2 site where the enzyme’s ATPase domain engages the nucleosome (Saha, et al., 2002; Saha, et al., 2005; Zofall, et al., 2006; Dechassa, et al., 2008) and that the isolated RSC catalytic subunit translocates naked DNA in ~2 bp increments (Sirinakis, et al., 2011). […] Rather, they may be compound steps composed of many 1-2 bp steps and may reflect relatively long kinetic pauses of DNA translocation imposed by the energy landscape of the nucleosomal substrates.”

*4) A prior EM structure of RSC complex by Leschziner and colleagues is highly supportive of the results in this paper and the model. In this structure, a large pocket resides within RSC complex, of nearly perfect dimensions of a nucleosome, and there is just enough room around the nucleosome in that pocket to also accommodate the propagation of a small DNA loop/wave of a few base pairs around the nucleosome, but not larger DNA wave/loop sizes, or the large unwrapping of DNA at the entry site. This existing structural support for the results and model would be worth mentioning in the Discussion.*

As suggested by the reviewers, we have discussed this paper and how it supports our findings (please see the second paragraph of the Discussion section).

*Specific editorial comments/requests (listed in order of appearance):*

*1) The authors observe two types of FRET traces, one with a decrease in FRET vs. one with an increase followed by a decrease, which they attribute to the two possible directions of RSC translocation, termed "entry-" and "exit-side movement". This interpretation is corroborated by control experiments in which a 2-nt gap has been introduced in each DNA strand, blocking each one direction of translocation. In the schematic describing these measurements (Figure 1, Figure 1—figure supplement 3), the gaps in DNA are located on opposite strands. Do SWI/SNF remodelers track a particular DNA strand? This is not discussed in the text and should be clarified.*

We have clarified this point by adding the following to the legend of Figure 1—figure supplement 3: “RSC translocates on DNA with a 3’→5’ polarity (Saha et al., 2005), and the gaps are positioned to prevent translocation in that direction. However, it has been previously shown that gaps in either strand prevent DNA translocation by RSC (Saha et al., 2005).”

*2) In Figure 1 (and elsewhere) the time trace shows the Cy3 fluorescence signal extinguish at ~80 s. Is the loss of signal due to dissociation of the nucleosome from DNA or is this photobleaching? Is there any evidence that nucleosomes dissociate from the DNA due to RSC and if so, is this observed for both entry- and exit-side movements? This is not discussed adequately.*

The loss of Cy3 fluorescence is primarily due to photobleaching. We do not expect to observe remodeler-induced nucleosome dissociation in these experiments because disassembly requires free DNA or nucleosomes as acceptors, the activity of histone chaperones, or a dinucleosome construct, all of which are absent in our single-molecule assay. In bulk experiments, other nucleosomes in solution or free DNA remaining after nucleosome reconstitution can act as acceptors for dimer or octamer transfer. In contrast, no such acceptors are available in our experiment as the nucleosomes are immobilized on the microscope slide and any free nucleosomes or DNA are washed away from the imaging chamber prior to the addition of RSC and ATP.

Consistent with these arguments, we do not see evidence that a substantial number of nucleosomes disassemble during the experiment. After incubating H2A/[end, +6] nucleosomes with 5 nM RSC and 5 µM ATP for 20 min, 91% of the nucleosomes remained attached to the glass slide with both DNA and octamer present. This fraction is similar to the fraction of nucleosomes remaining (91%) after buffer exchange and incubation with a mock buffer lacking RSC and ATP. Thus, while a very small fraction of nucleosomes may spontaneously disassemble in our assay, we see no evidence for RSC-mediated nucleosome disassembly in our single-molecule assay. We have clarified this point in the subsection “Probing for potential H2A-H2B dimer movement during remodeling”.

*3) In the text (Results) the authors state that the "[2-nt] gap at SHL+2, eliminated the zero FRET population and resulted in a single population with a FRET of ~0.14 after remodeling". However, there appears to be a small peak at 0 FRET in*
Figure 1*. What is this due to? According to the text, the SHL+2 gap site should favor "entry-side" motion, which should not generate a zero-FRET state.*

To build the histograms in Figure 1, we identified the initial (before remodeling) and final (after remodeling) FRET states from nucleosome remodeling traces. In a small subset of the traces from the nucleosomes with a gap at SHL+2, the Cy5 photobleached prior to the completion of remodeling, resulting in an apparent final zero FRET state. Although these traces end in a zero FRET state, the majority of them show the transient FRET increase characteristic of entry-side movement traces. Thus, these traces do not represent exit-side movement traces but rather entry-side movement traces where the Cy5 dye has bleached during remodeling. Furthermore, only ~8% of traces observed show a final FRET of zero, indicating that the vast majority of traces show the expected final FRET value. We have clarified this point in the subsection “Single-molecule FRET assay for monitoring nucleosome translocation by RSC”.

Although the zero FRET state seen in the exit-side movement traces could also potentially result from photobleaching of the Cy5 dye, we have done control experiments where we monitored remodeling of the H2A/[end, +6] nucleosomes with alternating 532 nm illumination (to excite the donor dye Cy3 and monitor the FRET value) and 647 nm illumination (to directly excite the acceptor dye Cy5 and check whether the Cy5 is bleached). The vast majority of the exit-side remodeling traces from these experiments showed that the final FRET state of exit-side movement represented a true zero FRET state and did not result from photobleaching of Cy5.

*4) In Figure 2, the histogram for FRET labeled nucleosomes in the presence of RSC, but no ATP (in red) seems to be bimodal, with a FRET state at ~0.3 and ~0.55, but this is not discussed in the text. Is this again attributable to photophysical effects, as rationalized for the small decrease in FRET with the addition of RSC? Or is this evidence for multiple conformations?*

We are not certain whether the ~0.3 FRET state is due to photophysical effect or a real FRET change. However, these events appear to be relatively rare (accounting for ~10% of the total events). Furthermore, they are not observed when RSC is added in the presence of ATP. For these reasons, we did not further investigate this state.

*5) In the text (Results) the authors claim that "If RSC were to unpeel a substantial amount of DNA and lift the DNA off the nucleosomal surface by a large distance, we would expect the FRET traces to exhibit a FRET decrease as the Cy5 dye on the linker DNA moves away from the nucleosome where the Cy3 dye resides". This statement is not quite as obvious as stated. This claim seems highly dependent on the geometry of the DNA-nucleosome complex (e.g. the locations of the nucleosome and DNA fluorescence labels) and seems to assume that the unpeeled DNA is fully extended. One could imagine situations where the unpeeled DNA contributes very little to the FRET or could even increase FRET, especially if the DNA end can bend (or loop) back on itself. This will obviously depend on the length / flexibility of DNA. As this is an important point of the manuscript that affects subsequent data interpretation, the authors should devote more effort to clarifying it and supporting it with more quantitative arguments.*

*On a related note, the authors repeatedly refer to "small-" and "large-" distance lifting of DNA of the nucleosome surface, but some numbers should be provided here. Although they provide an estimated bound of 0.5 nm from FRET, this seems to refer to inter-dye distance, which is not necessarily the same as the contour length of DNA lifted (especially if bending/looping is considered).* Based on the crystal structure of the nucleosome, breaking the outermost histone-DNA contacts at SHL ± 6.5 should induce a ~ 1 nm motion of the DNA end from the histone H2A labeling site in the H2A/[end, +6] construct, so unwrapping of DNA from the extreme ends of the nucleosome should be visible. However, this estimate indeed assumes that the unwrapped DNA stays fully extended, which is reasonable for unconstrained DNA given the ~ 150 bp persistence length of DNA. Nevertheless, because protein-DNA interactions can substantially alter the flexibility of DNA, we agree with the reviewers that we cannot exclude models where the remodeler has disrupted a substantially number of histone-DNA contacts simultaneously but still holds the unpeeled DNA near the octamer surface at the entry/exit sites of the nucleosome. We have mentioned this possibility in our original manuscript and have now made this point clearer in the revised manuscript, both in the Results section (subsection “Probing for potential DNA unwrapping at the edge of the nucleosome”, last paragraph) and in the Discussion section (second paragraph), as we also detailed in our response to the first part of comment 1.

We also note that regardless of whether the histone-DNA interactions remain intact, our data are consistent with movement of the Cy5-labeled DNA end near its canonical wrapping path, justifying our interpretation of the enzyme step size. Moreover, even if the movement of the DNA deviates from the canonical path, the small FRET changes that we observed in each step (~0.08 – 0.11) are consistent with the distance changes (~0.3 – 0.6 nm) expected for 1-2 bp of translocation, as we discussed in the third paragraph of the Discussion. These estimates are fairly robust to changes in the path of the DNA, and unwrapping of the DNA from the edge of the nucleosome (assuming no sharp kinks in the unwrapped DNA) does not substantially change the estimate of 0.3 nm/bp of translocation.

*6) The authors used a step-finding algorithm to determine the translocation step size of RSC. Were any selections made for the steps in their analysis or were all steps detected (regardless of size) shown in the histograms (Figure 4 and Figure 5)? There is a concern that such an algorithm could bias the step size distribution, by not reliably detecting smaller step sizes for instance. Did the authors test out other algorithms (e.g. t-test based step-finding, etc.) or try a pairwise distribution? If so, was the step size robust against detection method?*

We analyzed all steps detected by the step finding algorithm, and rejected steps only if 1) it involved a pause shorter than five frames or 2) the pauses occurred at a FRET value outside the range defined by our calibration curves where FRET was observed to vary linearly with distance. We added these criteria explicitly to the Materials and methods section (subsection “Data analysis”, second paragraph).

We were also concerned that the chi-square-minimization-based step-finding algorithm that we used (Kerssemakers, et al., 2006) could bias the step size distribution by not reliably detecting smaller step sizes. To test for this possibility, we simulated FRET traces that exhibited a ΔFRET step size of 0.053 with a similar amount of noise and distribution of pause lifetimes as experimentally detected. Analyzing this data produced an estimated ΔFRET of 0.058, indicating that our analysis is capable of detecting steps smaller than the ~0.1 ΔFRET steps observed.

In light of this comment, we performed additional analysis by reanalyzing the traces from the H2A/[end, +6] nucleosomes using an alternative step-finding algorithm based on a hidden Markov model (HMM) (McKinney et al., 2006). The HMM algorithm produced a similar step size distribution and an estimated step size of 1.9 ± 0.2 bp, similar to the 2.0 ± 0.2 bp estimate obtained using the chi-square-minimization-based algorithm. These new analysis results are presented in Figure 4—figure supplement 1.